# ADVERSARIAL POLICY OPTIMIZATION FOR OFFLINE PREFERENCE-BASED REINFORCEMENT LEARNING

**Hyungkyu Kang**
Seoul National University
Seoul, South Korea
hyungkyu0119@snu.ac.kr

**Min-hwan Oh**
Seoul National University
Seoul, South Korea
minoh@snu.ac.kr

## ABSTRACT

In this paper, we study offline preference-based reinforcement learning (PbRL), where learning is based on pre-collected preference feedback over pairs of trajectories. While offline PbRL has demonstrated remarkable empirical success, existing theoretical approaches face challenges in ensuring conservatism under uncertainty, requiring computationally intractable confidence set constructions. We address this limitation by proposing Adversarial Preference-based Policy Optimization (`APPO`), a computationally efficient algorithm for offline PbRL that guarantees sample complexity bounds without relying on explicit confidence sets. By framing PbRL as a two-player game between a policy and a model, our approach enforces conservatism in a tractable manner. Using standard assumptions on function approximation and bounded trajectory concentrability, we derive a sample complexity bound. To our knowledge, `APPO` is the first offline PbRL algorithm to offer both statistical efficiency and practical applicability. Experimental results on continuous control tasks demonstrate that `APPO` effectively learns from complex datasets, showing comparable performance with existing state-of-the-art methods.

## 1 INTRODUCTION

While Reinforcement Learning (RL) has achieved remarkable success in real-world applications (Mnih, 2013; Silver et al., 2017; Kalashnikov et al., 2018; Brohan et al., 2022), its performance heavily depends on the design of the reward function (Wirth et al., 2017), which can be challenging in practice. To address this issue, preference-based reinforcement learning (PbRL), also known as reinforcement learning with human feedback, has gained increasing attention as an alternative to manually designed rewards. In PbRL, a reward model is learned from preference feedback provided by human experts, who compare pairs of trajectories (Christiano et al., 2017). This approach enables the learning process to align better with human intentions. PbRL has demonstrated its effectiveness in various domains, including gaming (MacGlashan et al., 2017; Christiano et al., 2017; Warnell et al., 2018), natural language processing (Ziegler et al., 2019; Stiennon et al., 2020; Nakano et al., 2021; Ouyang et al., 2022; Bai et al., 2022), and robotics (Brown et al., 2019; Shin et al., 2023).

However, collecting preference feedback can be costly, especially when real-time feedback from human experts is required. In such cases, learning from pre-collected data is preferred over online learning. This approach is referred to as *offline* PbRL, where the learning process relies solely on pre-collected trajectories and preference feedback. Empirical studies have shown the effectiveness of offline PbRL (Kim et al., 2023; An et al., 2023; Shin et al., 2023; Hejna & Sadigh, 2024), leveraging techniques from deep RL literature. On the theoretical side, prior works prove that trajectory concentrability with respect to the data-collecting distribution leads to a sample complexity bound (Zhu et al., 2023; Zhan et al., 2024a; Pace et al., 2024). However, they rely on the explicit construction of confidence sets to achieve conservatism (pessimism). Dealing with such confidence sets in the general function approximation setting requires intractable optimizations: Zhan et al. (2024a) involve tri-level constrained optimization with respect to the confidence sets of rewards and transitions, Pace et al. (2024) use uncertainty penalty defined as the width of confidence sets, and the analysis of Zhu et al. (2023) is restricted to linear models. Despite provable sample complexity bounds, existing offline PbRL algorithms become computationally intractable with general function approximation.

In this work, we propose a computationally and statistically efficient offline PbRL algorithm, *Adversarial Preference-based Policy Optimization* (APPO). Our analysis is based on general function approximation for both the model and value function classes. Moreover, standard assumptions on function classes and bounded trajectory concentrability (Zhan et al., 2024a) are sufficient to establish our sample complexity bound. Beyond its strong statistical guarantees, our algorithm is simple to implement using standard optimization techniques. The core idea behind our algorithm is the two-player game formulation of model-based PbRL, which has been used in other areas of RL (Rajeswaran et al., 2020; Rigter et al., 2022; Cheng et al., 2022; Shen et al., 2024; Bhardwaj et al., 2024). By framing PbRL as a game between a policy and a model, we ensure conservatism without explicitly constructing intractable confidence sets. Furthermore, our novel reparameterization technique allows us to find a near-optimal policy efficiently via adversarial training. To the best of our knowledge, APPO is the first offline PbRL algorithm with both statistical performance guarantees and a practical implementation. Our contributions can be summarized as follows:

- We propose APPO, a simple algorithm for offline PbRL with general function approximation. Based on the two-player game formulation of PbRL in conjunction with our reparameterization technique for the reward model, our algorithm ensures provable conservatism without explicit construction of confidence sets. To our knowledge, our APPO is the first computationally efficient offline PbRL algorithm providing a sample complexity bound.

- We prove the sample complexity of our proposed algorithm under standard assumptions on the function classes and concentrability. The result is rooted in our novel sub-optimality decomposition, which shows that adversarial training leads to model conservatism.

- We present a practical implementation of APPO that can learn with large datasets using neural networks. Experiments on continuous control tasks demonstrate that APPO achieves performance comparable to existing state-of-the-art algorithms.

## 1.1 RELATED WORK

**Provable Online PbRL.** In the tabular setting, Novoseller et al. (2020) developed an algorithm grounded in posterior sampling and the dueling bandit framework (Yue et al., 2012), demonstrating an asymptotic rate for Bayesian regret. Xu et al. (2020) proposed an algorithm leveraging an exploration bonus for previously unseen states, which provides a sample complexity bound. Saha et al. (2023) and Zhan et al. (2024b) focused on the linear preference model with a known linear feature map, each offering regret and sample complexity bounds. However, their algorithms require solving an optimization $\arg\max_{\pi,\pi'} \|\mathbb{E}_{\tau\sim\pi}[\phi(\tau)] - \mathbb{E}_{\tau\sim\pi'}[\phi(\tau)]\|_\Sigma$ for some positive definite matrix $\Sigma$, which is computationally intractable. To address this challenge in the linear model, Wu & Sun (2024) devised a randomized algorithm with a provable regret bound and further proposed a model-based posterior sampling algorithm under the bounded Eluder dimension (Russo & Van Roy, 2013) assumption, ensuring bounded Bayesian regret. Recent works have also explored provably efficient algorithms under the general function approximation setting (Chen et al., 2022; Wu & Sun, 2024; Chen et al., 2023). Chen et al. (2022) introduced an exploration-bonus-based algorithm that provides bounded regret in both pairwise and n-wise comparison settings. Additionally, Chen et al. (2023) leveraged the conditional value-at-risk operator (Artzner, 1997) to devise an algorithm with a regret guarantee. Du et al. (2024) took a different approach, studying neural function approximation in the context of reward models. In another notable work, Swamy et al. (2024) reframed PbRL as a zero-sum game between two policies, encompassing general reward models.

**Provable Offline PbRL.** While there has been a growing amount of research on online PbRL, the theoretical understanding of offline PbRL remains relatively limited. A primary challenge in offline PbRL, much like in offline standard RL, is ensuring sufficient conservatism in the model. Zhu et al. (2023) addressed this challenge by proposing a pessimistic maximum likelihood estimation (MLE) algorithm for the linear model with known transitions. Zhan et al. (2024a) extended this idea to general function approximation, highlighting the importance of trajectory concentrability in establishing a lower bound for sample complexity. Despite the provable sample complexity bound, their proposed algorithm, FREEHAND-transition, relies on solving $\arg\max_\pi \arg\min_{r\in\hat{\mathcal{R}}} \arg\min_{P\in\hat{\mathcal{P}}}\{\mathbb{E}_{\tau\sim P,\pi}[r(\tau)] - \mathbb{E}_{\tau\sim P^\star,\pi}[r(\tau)]\}$ where $\hat{\mathcal{R}}$ is the confidence set of rewards and $\hat{\mathcal{P}}$ is the confidence set of transitions, which is intractable in practice. Pace et al. (2024) study a different but related setting, where the agent elicits high-quality

preference information from offline data. Their method achieves conservatism through explicit uncertainty penalties defined as $u_R(\tau) = \sup_{r_1, r_2 \in \hat{\mathcal{R}}} |r_1(\tau) - r_2(\tau)|$ (reward uncertainty) and $u_P(s, a) = \sup_{P_1, P_2 \in \hat{\mathcal{P}}} \|P_1(\cdot \mid s, a) - P_2(\cdot \mid s, a)\|_1$ (transition uncertainty), which makes it intractable with general function approximation. Chang et al. (2024) also explored a slightly different scenario where the data collection policy is known and online interaction is allowed. They demonstrated that a simple natural policy gradient combined with MLE reward is provably efficient, but their sample complexity bound is affected by an additional concentrability coefficient relative to KL-regularized policies.

**Adversarial Training in RL.** Adversarial training is a widely used approach in RL literature (Rajeswaran et al., 2020; Pásztor et al., 2024), especially offline (standard) RL (Rigter et al., 2022; Cheng et al., 2022; Bhardwaj et al., 2024). The basic idea is to leverage adversarial training to implement conservative policy optimization. Recently, adversarial training has also been applied in human preference alignment (Makar-Limanov et al., 2024; Cheng et al., 2024; Shen et al., 2024). The most closely related work to ours is Shen et al. (2024), which also formulated PbRL as a two-player game. However, their focus is on online PbRL, and while they provide proof of convergence for the optimization objective, this does not necessarily translate into a sample complexity guarantee.

## 2 PRELIMINARIES

**Markov Decision Processes.** We consider an episodic MDP $(\mathcal{S}, \mathcal{A}, H, \{P_h^\star\}_{h=1}^H, \{r_h^\star\}_{h=1}^H)$, where $\mathcal{S}$ and $\mathcal{A}$ are the state space and the action space, $H$ is the length of each episode, $P^\star = \{P_h^\star\}_{h=1}^H$ is the collection of transition probability distributions, and $r^\star = \{r_h^\star\}_{h=1}^H$ is the collection of reward functions. Each episode starts at some initial state $s_1$ without loss of generality[1], and the episode ends after $H$ steps. For each step $h \in [H]$, the agent observes the state $s_h$, and then takes action $a_h$. The environment generates reward $r_h^\star(s_h, a_h)$ (note that, in the preference-based learning setting, rewards at each step are unobservable to the agent) and next state $s_{h+1}$ according to the transition probability $P_h^\star(\cdot \mid s_h, a_h)$.

The agent takes actions based on its policy $\pi = \{\pi_h\}_{h \in [H]}$, where $\pi_h(\cdot \mid s)$ is a probability distribution over $\mathcal{A}$. The state-value function and the action-value function of policy $\pi$ with respect to reward $r = \{r_h\}_{h=1}^H$ are the expected sum of rewards up to termination, starting from $s_h = s$ and $(s_h, a_h) = (s, a)$ respectively, following the policy $\pi$. Formally, they are defined as

$$V_{h,r}^\pi(s) := \mathbb{E}_\pi \left[ \sum_{h'=h}^H r_h(s_{h'}, a_{h'}) \mid s_h = s \right], \quad Q_{h,r}^\pi := \mathbb{E}_\pi \left[ \sum_{h'=h}^H r_h(s_{h'}, a_{h'}) \mid s_h = s, a_h = a \right].$$

To simplify the notation, for $g : \mathcal{S} \mapsto \mathbb{R}$, we use $Pg(s, a)$ to denote $\mathbb{E}_{s' \sim P(\cdot \mid s, a)}[g(s')]$. For any policy $\pi$ and reward $r$, the Bellman equation relates $Q^\pi$ to $V^\pi$ as

$$Q_{h,r}^\pi(s, a) = r_h(s, a) + P^\star V_{h+1,r}^\pi(s, a), \quad V_{h,r}^\pi(s) = \mathbb{E}_{a \sim \pi_h(\cdot \mid s)}[Q_{h,r}^\pi(s, a)], \quad V_{H+1}^\pi(s) = 0.$$

Given a policy $\pi = \{\pi_h\}_{h \in [H]}$, we define the state visitation distribution as $d_h^\pi(s) := \mathbb{P}_\pi(s_h = s)$ where $\mathbb{P}_\pi$ is the probability distribution of trajectories $(s_1, a_1, \ldots, s_H, a_H)$ when the agent uses policy $\pi$. We overload the notation to denote the state-action visitation distribution, $d_h^\pi(s, a) := \mathbb{P}_\pi(s_h = s, a_h = a)$. In addition, we denote the distribution of trajectories under $\pi$ by $d^\pi(\tau)$.

**Offline Preference-based Reinforcement Learning.** We consider the offline PbRL problem, where the agent cannot observe the true reward $r^\star$ but only binary preference feedback over trajectory pairs. Specifically, we are given a preference dataset $\mathcal{D}_{\text{pref}} = \{(\tau^{m,0}, \tau^{m,1}, y^m)\}_{m=1}^M$ that consists of i.i.d. trajectory pairs $\tau^{m,i} = \{s_h^{m,i}, a_h^{m,i}\}_{h=1}^H$ ($i = 0, 1$) sampled by some reference policy $\mu$. For a monotonically increasing link function $\Phi : \mathbb{R} \mapsto [0, 1]$, we assume the preference feedback $y^m \in \{0, 1\}$ is generated by the following preference model:

$$\mathbb{P}(y = 1 \mid \tau^0, \tau^1) = \mathbb{P}(\tau^1 \text{ is preferred over } \tau^0) = \Phi(r^\star(\tau^1) - r^\star(\tau^0))$$

where we denote $r^\star(\tau) = \sum_{h=1}^H r_h^\star(s_h, a_h)$ for given trajectory $\tau = (s_1, a_1, \ldots, s_H, a_H)$. Additionally, we assume that $\kappa = 1/(\inf_{x \in [-R,R]} \Phi'(x))$, where $R$ is a bound on trajectory returns, is

---

[1]Our result easily extends to the general case with an initial distribution $\rho(\cdot)$. We can modify the MDP by setting a fixed initial state $s_1$ and $\mathbb{P}_1(\cdot \mid s_1, a) = \rho(\cdot)$ for all $a \in \mathcal{A}$.

finite. When $\Phi$ is set to be the sigmoid function $\sigma(x) = 1/(1 + \exp(-x))$, we obtain the widely used Bradely-Terry-Luce model (Bradley & Terry, 1952). In addition to the preference dataset, we have an unlabeled trajectory dataset $\mathcal{D}_{\text{traj}} = \{(\tau^{n,0}, \tau^{n,1})\}_{n=1}^N$ where the trajectory pairs are sampled i.i.d. by executing the reference policy $\mu$. The agent's goal is to find an $\epsilon$-optimal policy $\hat{\pi}$ with respect to the optimal policy $\pi^\star$, which satisfies $V_{1,r^\star}^{\pi^\star}(s_1) - V_{1,r^\star}^{\hat{\pi}}(s_1) \leq \epsilon$.

**General Function Approximation.** We consider general function approximation for rewards and transitions: the function class of rewards $\mathcal{R}$ and the function class of transitions $\mathcal{P}$. We do not impose any specific structure on them, so $\mathcal{R}$ and $\mathcal{P}$ can contain expressive functions such as neural networks. Based on the function classes, we construct a reward model using maximum likelihood estimation $\hat{r} \in \arg\min_{r \in \mathcal{R}^H} \hat{\mathcal{L}}_R(r)$ where

$$\hat{\mathcal{L}}_R(r) = -\mathop{\mathbb{E}}_{(\tau^0, \tau^1, y) \sim \mathcal{D}_{\text{pref}}} \left[ y \cdot \log \Phi(r(\tau^1) - r(\tau^0)) + (1 - y) \cdot \log \Phi(r(\tau^0) - r(\tau^1)) \right].$$

Similarly, we learn a transition model $\hat{P}_h \in \arg\min_{P \in \mathcal{P}} \hat{\mathcal{L}}_T(P; h)$ for all $h \in [H]$, where

$$\hat{\mathcal{L}}_T(P; h) = \mathbb{E}_{(s_h, a_h, s_{h+1}) \sim \mathcal{D}_{\text{traj}}} [\log P(s_{h+1} \mid s_h, a_h)]$$

**Additional Notations.** We denote $[n] := \{1, 2, \ldots, n\}$ for $n \in \mathbb{N}$. For $x, y \in \mathbb{R}^d$, $\langle x, y \rangle$ denotes the inner product of $x$ and $y$. Given a function $f : \mathcal{S} \times \mathcal{A} \mapsto \mathbb{R}$ and a policy $\pi$, we write $f \circ \pi(s) := \mathbb{E}_{a \sim \pi(\cdot|s)}[f(s, a)]$. For given dataset $\mathcal{D}$, we use $\mathbb{E}_{x \sim \mathcal{D}}[f(x)]$ to denote $\frac{1}{|\mathcal{D}|} \sum_{x \in \mathcal{D}} f(x)$.

## 3 ALGORITHM

### 3.1 PbRL AS A TWO-PLAYER GAME

A previous study on model-based PbRL by Zhan et al. (2024a) proves that the following optimization problem yields a near-optimal policy $\hat{\pi}$, for an appropriately chosen constant $\zeta$:

$$\hat{\pi} \in \arg\max_{\pi} \min_{r \in \hat{\mathcal{R}}} \left( V_{1,r}^{\pi}(s_1) - V_{1,r}^{\mu}(s_1) \right) \text{ where } \hat{\mathcal{R}} = \left\{ r \in \mathcal{R}^H : \hat{\mathcal{L}}_R(r) \leq \hat{\mathcal{L}}_R(\hat{r}) + \zeta \right\}. \quad (1)$$

The minimization with respect to reward model $r \in \hat{\mathcal{R}}$ ensures conservatism, which is essential for a provable guarantee. However, the constrained optimization is intractable with general function approximation. To address this challenge, we formulate the model-based PbRL problem as a two-player Stackelberg game (Von Stackelberg, 2010) between the policy and the reward:

$$\hat{\pi} \in \arg\max_{\pi} \left( V_{1,r^\pi}^{\pi}(s_1) - V_{1,r^\pi}^{\mu}(s_1) \right)$$

$$\text{subject to } r^\pi \in \arg\min_{r \in \mathcal{R}^H} \left( V_{1,r}^{\pi}(s_1) - V_{1,r}^{\mu}(s_1) + \mathcal{E}(r; \hat{r}) \right). \quad (2)$$

Here, $\mathcal{E}(r; \hat{r})$ is a loss function that penalizes $r$ if it deviates from $\hat{r}$. In the Stackelberg game formulation, the reward minimizes $V_{1,r}^{\pi}(s_1) - V_{1,r}^{\mu}(s_1)$, while the policy maximizes it. We can interpret this competition by viewing $V_{1,r}^{\pi}(s_1) - V_{1,r}^{\mu}(s_1)$ as the relative performance of $\pi$ compared to $\mu$ with respect to reward $r$. Intuitively, $\pi$ maximizes the cumulative reward $r^\pi$, as in the standard RL setup. However, $r^\pi$ minimizes the cumulative reward when evaluated under $\pi$. This competition facilitates conservatism and makes $\pi$ robust to model error.

Then, what loss function $\mathcal{E}$ leads to a provable bound? A naive choice might be $\hat{\mathcal{L}}_R(r) - \hat{\mathcal{L}}_R(\hat{r})$, as it leads to the Lagrangian dual form of the optimization problem in (1), while disregarding the Lagrangian multiplier. However, the loss $\mathcal{E}(r; \hat{r}) = \hat{\mathcal{L}}_R(r) - \hat{\mathcal{L}}_R(\hat{r})$ does not guarantee statistical efficiency, because the Stackelberg game in (2) does not include the Lagrangian multiplier for the likelihood constraint. Instead, we propose the trajectory-pair $\ell_1$ loss:

$$\mathcal{E}(r; \hat{r}) = \mathbb{E}_{\tau^0, \tau^1 \sim \mu} \left[ \left| \{r(\tau^0) - r(\tau^1)\} - \{\hat{r}(\tau^0) - \hat{r}(\tau^1)\} \right| \right],$$

which leads to a provable guarantee (Theorem 4.1). Intuitively, this loss measures the deviation of $r$ from $\hat{r}$ by evaluating the difference in total reward (return) between the two trajectories. Given the unlabeled trajectory dataset $\mathcal{D}_{\text{traj}}$, we approximate $\mathcal{E}(r; \hat{r})$ with its finite-sample version:

$$\hat{\mathcal{E}}_{\mathcal{D}_{\text{traj}}}(r; \hat{r}) = \mathbb{E}_{(\tau^0, \tau^1) \sim \mathcal{D}_{\text{traj}}} \left[ \left| \{r(\tau^0) - r(\tau^1)\} - \{\hat{r}(\tau^0) - \hat{r}(\tau^1)\} \right| \right].$$

In the following two sections, we discuss how to implement the optimization in (2) in a sample-efficient manner.

---

**Algorithm 1** Adversarial Preference-based Policy Optimization with Rollout (`APPO-rollout`)

1: **Input:** Number of rollouts $K_1, K_2$, constant $\eta$, $\pi_h^1 = \text{Unif}(\mathcal{A})$ for all $h \in [H]$
2: Estimate $\hat{r} \in \arg\min_{r \in \mathcal{R}^H} \hat{\mathcal{L}}_R(r)$
3: **for** $t = 1, \cdots, T$ **do**
4:      Execute $\pi^t$ to collect $K_1$ trajectories $\mathcal{D}_{\text{rollout}}^t$
5:      Optimize $r^t \in \arg\min_{r \in \mathcal{R}^H} \left( \mathbb{E}_{\tau \sim \mathcal{D}_{\text{rollout}}^t}[r(\tau)] - \mathbb{E}_{\tau \sim \mathcal{D}_{\text{traj}}}[r(\tau)] + \lambda \hat{\mathcal{E}}_{\mathcal{D}_{\text{traj}}}(r; \hat{r}) \right)$
6:      Compute $\bar{Q}^t$ via $\text{PE}(\mu, \pi^t, \hat{r}, K_2)$ in Algorithm 3
7:      Update policy $\pi_h^{t+1}(a \mid s) \propto \pi_h^t(a \mid s) \exp(\eta \bar{Q}_h^t(s, a))$ for all $h \in [H]$
8: **end for**
9: Return $\bar{\pi} = \frac{1}{T} \sum_{t=1}^T \pi_t$

---

**Algorithm 2** Adversarial Preference-based Policy Optimization (`APPO`)

1: **Input:** Constant $\eta$, Initial policy $\pi_h^1 = \text{Unif}(\mathcal{A})$ for all $h \in [H]$
2: Estimate $\hat{r} \in \arg\min_{r \in \mathcal{R}^H} \hat{\mathcal{L}}_R(r)$, $\hat{P}_h \in \arg\min_{P \in \mathcal{P}} \hat{\mathcal{L}}_T(P; h)$ for all $h \in [H]$
3: **for** $t = 1, \cdots, T$ **do**
4:      $f^t \in \arg\min_{f \in \mathcal{F}^H} \left( \sum_{h=1}^H \mathbb{E}_{(s_h, a_h) \sim \mathcal{D}_{\text{traj}}} [f_h \circ \pi_h^t(s_h) - f_h(s_h, a_h)] + \lambda \hat{\mathcal{E}}_{\mathcal{D}_{\text{traj}}}(f; \hat{P}, \hat{r}) \right)$
5:      Update policy $\pi_h^{t+1}(a \mid s) \propto \pi_h^t(a \mid s) \exp(\eta f_h^t(s, a))$ for $h \in [H]$
6: **end for**
7: Return $\hat{\pi} = \frac{1}{T} \sum_{t=1}^T \pi^t$

---

## 3.2 Adversarial Optimization for PbRL

In this section, we present an algorithm, `APPO-rollout`, that serves as a building block of our main algorithm. For `APPO-rollout`, we consider the setting where the transition $P^\star$ is known or where online interaction (without preference feedback) is possible. This is a temporary assumption, as our main algorithm (Algorithm 2) works with an unknown transition.

Algorithm 1 presents the pseudo-code of `APPO-rollout`, which is based on the Stackelberg game formulation of PbRL that we discussed. Inspired by the adversarial training methods in offline RL under the standard setting (Cheng et al., 2022; Rigter et al., 2022; Bhardwaj et al., 2024), we alternately optimize the policy and reward to solve the optimization problem in (2).

**Reward Model Update for Provable Conservatism.** The reward model update aims to solve the following optimization problem approximately:

$$\arg\min_{r \in \mathcal{R}^H} \left( \mathbb{E}_{\tau \sim \pi^t}[r(\tau)] - \mathbb{E}_{\tau \sim \mu}[r(\tau)] + \lambda \mathcal{E}(r; \hat{r}) \right) = \arg\min_{r \in \mathcal{R}^H} \left( V_{1,r}^{\pi^t}(s_1) - V_{1,r}^{\mu}(s_1) + \lambda \mathcal{E}(r; \hat{r}) \right), \quad (3)$$

which is the inner optimization in (2). The expectations $\mathbb{E}_{\tau \sim \mu}[r(\tau)]$ and $\mathcal{E}(r; \hat{r})$ are approximated using offline data $\mathcal{D}_{\text{traj}}$. Also, we collect trajectories by executing $\pi^t$, to compute the finite-sample version of $\mathbb{E}_{\tau \sim \pi^t}[r(\tau)]$. Note that the trajectory rollout (Line 4) is possible since we assume a known transition $P^\star$ or access to online interaction.

**Policy Update.** After optimizing $r^t$, we estimate the action-value function of $\pi^t$ with respect to $r^t$ using a policy evaluation subroutine PE, whose pseudo-code is provided in Algorithm 3. This subroutine computes an approximate value function $\bar{Q}^t$ using Monte Carlo estimation, providing an error bound relative to the true value function $Q_{r^t}^{\pi^t}$. The theoretical analysis of PE is presented in Appendix B. With the estimated value function $\bar{Q}^t$, we then proceed to update the policy using trust region policy optimization (TRPO) (Schulman et al., 2015) update.

## 3.3 `APPO`: Reparameterized Algorithm for Unknown Transition

In this section, we consider the setting where the transition $P^\star$ is unknown. In Algorithm 1, the information from the transition $P^\star$ is utilized in Line 4, where we collect on-policy trajectories to approximate $\mathbb{E}_{\tau \sim \pi^t}[r(\tau)]$. Moreover, the policy evaluation step in Algorithm 3 involves trajectory

rollouts. To bypass such on-policy rollouts, we make the following observation:

$$\mathbb{E}_{\tau \sim \pi^t}[r(\tau)] - \mathbb{E}_{\tau \sim \mu}[r(\tau)] = V_{1,r}^{\pi^t}(s_1) - V_{1,r}^{\mu}(s_1)$$

$$= \sum_{h=1}^{H} \mathbb{E}_{(s_h, a_h) \sim d_h^{\mu}} \left[ (Q_{h,r}^{\pi^t} \circ \pi_h^t)(s_h) - Q_{h,r}^{\pi^t}(s_h, a_h) \right] \quad (4)$$

which is due to the performance difference lemma (Lemma E.1). Since the expectation on the right is taken with respect to $d_h^{\mu}$, the data-generating distribution of $\mathcal{D}_{\text{traj}}$, we can approximate it using $\mathcal{D}_{\text{traj}}$. Furthermore, given the policy $\pi^t$, the Bellman equation implies a mapping between reward models and action-value functions. Specifically, for given reward model $r = \{r_h\}_{h=1}^{H}$, we have the action-value function $\{Q_{h,r}^{\pi^t}\}_{h=1}^{H}$. Conversely, suppose that we have a function class $\mathcal{F}$ that contains every action-value function. For $f = \{f_h\}_{h=1}^{H} \in \mathcal{F}^H$, we can construct the corresponding reward model satisfying the Bellman equation $f_h = r_h + P_h^{\star}(f_{h+1} \circ \pi_{h+1}^t)$. Formally, we define the induced reward models:

**Definition 1** (Induced reward model). *Given $f = \{f_h\}_{h=1}^{H} \in \mathcal{F}^H$, and a policy $\{\pi_h\}_{h=1}^{H}$, we define the induced reward model $r_{P^{\star}, f}^{\pi} = \{r_{h,P^{\star},f}^{\pi}\}_{h=1}^{H}$ where $r_{h,P^{\star},f}^{\pi} = f_h - P_h^{\star}(f_{h+1} \circ \pi_{h+1})$ for $h \in [H]$ (we set $f_{H+1} = 0$ by convention).*

Therefore, given reward model $r$ and action-value function $f$, we have that

$$Q_{h,r}^{\pi} = r_h + P^{\star}(Q_{h+1,r}^{\pi} \circ \pi_{h+1}), \quad f_h = r_{h,P^{\star},r}^{\pi} + P_h^{\star}(f_{h+1} \circ \pi_h) \text{ for all } h \in [H].$$

Importantly, *the mapping does not need to be bijective* for our theoretical analysis, as long as the Bellman equation holds. Using this mapping in conjunction with our observation in (4), we reparameterize the optimization problem in (3) as:

$$\underset{f \in \mathcal{F}^H}{\arg \min} \left( \sum_{h=1}^{H} \mathbb{E}_{(s_h, a_h) \sim d_h^{\mu}} \left[ (f_h \circ \pi_h^t)(s_h) - f_h(s_h, a_h) \right] + \lambda \mathcal{E}(f; P^{\star}, \hat{r}) \right) \quad (5)$$

$$\text{where } \mathcal{E}(f; P^{\star}, \hat{r}) = \mathbb{E}_{(\tau_0, \tau_1) \sim \mu} \left[ \left| \{ r_{P^{\star},f}^{\pi^t}(\tau^0) - r_{P^{\star},f}^{\pi^t}(\tau^1) \} - \{ \hat{r}(\tau^0) - \hat{r}(\tau^1) \} \right| \right].$$

The offline dataset $\mathcal{D}_{\text{traj}}$ is sufficient to approximate the optimization objective in (5) with

$$\mathbb{E}_{(s_h, a_h) \sim \mathcal{D}_{\text{traj}}} \left[ (f_h \circ \pi_h^t)(s_h) - f_h(s_h, a_h) \right] \approx \mathbb{E}_{(s_h, a_h) \sim d_h^{\mu}} \left[ (f_h \circ \pi_h^t)(s_h) - f_h(s_h, a_h) \right]$$

$$\hat{\mathcal{E}}_{\mathcal{D}_{\text{traj}}}(f; \hat{P}, \hat{r}) := \mathbb{E}_{(\tau_0, \tau_1) \sim \mathcal{D}_{\text{traj}}} \left[ \left| \{ r_{\hat{P},f}^{\pi^t}(\tau^0) - r_{\hat{P},f}^{\pi^t}(\tau^1) \} - \{ \hat{r}(\tau^0) - \hat{r}(\tau^1) \} \right| \right] \approx \mathcal{E}(f; P^{\star}, \hat{r}),$$

where we use the estimated transition model $\hat{P}$ in place of $P^{\star}$. Moreover, since we directly optimize the action-value function, a policy evaluation oracle is not required to update the policy. Therefore, this reparameterization allows us to solve the optimization problem in (2) without access to the true transition $P^{\star}$ or a policy evaluation oracle. The complete pseudo-code is presented in Algorithm 2.

**Remark on Computational Complexity.** The computational complexity of APPO is primarily determined by the value function optimization (Line 4) and the policy update (Line 5). Although optimizing $f^t$ is generally a non-convex problem, it can be efficiently implemented using gradient-based methods when $\mathcal{F}$ is a class of neural networks. For the policy update, it is known that $\pi_h^{t+1}(a \mid s) \propto \pi_h^t(a \mid s) \exp(\eta f_h^t(s, a))$ is derived from the TRPO objective (Schulman et al., 2015; Neu et al., 2017):

$$\pi_h^{t+1} \in \arg \max_{\pi} \mathbb{E}_{s_h \sim d_h^{\pi^t}} \left[ f_h^t \circ \pi(s_h) - \eta^{-1} D_{KL} \left( \pi(\cdot \mid s_h) \| \pi_h^t(\cdot \mid s_h) \right) \right],$$

which is widely used in deep RL. As a result, the policy update is efficient within the deep learning framework. In practice, other policy optimization techniques (Schulman et al., 2017; Fujimoto et al., 2018; Haarnoja et al., 2018) can also be applied. Overall, APPO relies on solving two standard non-convex optimizations to compute $f^t$ and $\pi^t$, both of which are practical to implement with neural function approximation. This computational efficiency contrasts with that of existing offline PbRL algorithms, which require intractable optimization over confidence sets, as discussed in Section 1.1.

## 4 THEORETICAL ANALYSIS

In this section, we present theoretical analyses of our proposed algorithm, `APPO`. We note that `APPO-rollout` also guarantees a sample complexity bound, which is presented in Appendix C.

We assume the reward class $\mathcal{R}$ and the transition class $\mathcal{P}$ are realizable and rewards are bounded. These are standard assumptions (Chen et al., 2023; Zhan et al., 2024a; Pace et al., 2024).

**Assumption 1** (Reward realizability). *We have $r_h^\star \in \mathcal{R}$ for all $h \in [H]$. In addition, every $r \in \mathcal{R}^H$ satisfies $0 \leq r(\tau) \leq R$ for any trajectory $\tau$.*

**Assumption 2** (Transition realizability). *We have $P_h^\star \in \mathcal{P}$ for all $h \in [H]$.*

Additionally, we introduce the value function class and assume it is bounded. Note that every $Q_{h,r}^\pi$ satisfies the condition $\|f\|_\infty \leq R$ due to Assumption 1.

**Assumption 3** (Value function class). *For any $h \in [H]$, $r \in \mathcal{R}^H$, and policy $\pi$, we have $Q_{h,r}^\pi \in \mathcal{F}$. In addition, every $f \in \mathcal{F}$ satisfies $0 \leq f(s, a) \leq R$ for all $(s, a) \in \mathcal{S} \times \mathcal{A}$.*

The following assumption defines the trajectory concentrability coefficient between the optimal policy $\pi^\star$ and the reference policy $\mu$.

**Assumption 4** (Trajectory concentrability). *There exists a finite constant $C_{TR}$ such that the behavior policy $\mu$ and the optimal policy $\pi^\star$ satisfy $\sup_\tau \frac{d^{\pi^\star}(\tau)}{d^\mu(\tau)} \leq C_{TR}$.*

The bounded $C_{\text{TR}}$ ensures that the support of $d^\mu$ sufficiently covers the support of $d^{\pi^\star}$, similar to the concentrability condition in Zhan et al. (2024a)[2]. As a result, we expect $\mathcal{D}_{\text{traj}}$ to contain high-quality trajectories. The lower bound in Zhan et al. (2024a) shows that the trajectory concentrability is essential in offline PbRL. This implies that offline PbRL is strictly more challenging than offline standard RL, where step-wise concentrability is sufficient to achieve a performance guarantee (Uehara & Sun, 2022). We now present the sample complexity bound.

**Theorem 4.1.** *Suppose Assumptions 1,2, 3, and 4 hold. With probability at least $1 - \delta$, Algorithm 2 with $\lambda = \Theta(C_{TR}), \lambda > C_{TR}, \eta = \sqrt{\frac{2 \log |\mathcal{A}|}{R^2 T}}$ achieves*

$$V_{1,r^\star}^{\pi^\star} - V_{1,r^\star}^{\hat{\pi}}$$
$$\leq \mathcal{O}\left( C_{TR} \sqrt{\frac{\kappa^2 H}{M} \log \frac{|\mathcal{R}|}{\delta}} + RH \sqrt{\frac{1}{N} \max\left\{ HT \log \frac{H|\mathcal{F}|}{\delta}, \log \frac{H|\mathcal{P}|}{\delta} \right\}} + RH \sqrt{\frac{\log |\mathcal{A}|}{T}} \right).$$

*Setting $T = \Theta\left( \frac{R^2 H^2 \log |\mathcal{A}|}{\epsilon^2} \right)$, $N = \Theta\left( \max\left\{ \frac{R^4 H^5 \log |\mathcal{A}| \log(H|\mathcal{F}|/\delta)}{\epsilon^4}, \frac{R^2 H^2 \log(H|\mathcal{P}|/\delta)}{\epsilon^2} \right\} \right)$, and $M = \Theta\left( \frac{C_{TR}^2 \kappa^2 H \log(|\mathcal{R}|/\delta)}{\epsilon^2} \right)$, Algorithm 2 achieves $\epsilon$-optimal policy, i.e. $V_{1,r^\star}^{\pi^\star} - V_{1,r^\star}^{\hat{\pi}} \leq \epsilon$.*

**Discussion on Theorem 4.1.** Our analysis naturally extends to infinite function classes by applying the standard covering number argument, replacing the cardinalities $|\mathcal{R}|$, $|\mathcal{P}|$, and $|\mathcal{F}|$ with covering numbers. To our knowledge, FREEHAND-transition Zhan et al. (2024a) is the only statistically efficient algorithm for offline PbRL in stochastic MDPs. Our sample complexity bound matches theirs for labeled data $(M)$. However, FREEHAND-transition requires $\Theta\left( \frac{C_P^2 R^2 H^2 \log(H|\mathcal{P}|/\delta)}{\epsilon^2} \right)$ unlabeled trajectories where $C_P$ is the trajectory concentrability for transition [3]. This highlights a trade-off: While FREEHAND-transition has tighter bounds for unlabeled data $(N)$, it is computationally intractable. That is, FREEHAND-transition requires solving a nearly intractable nested optimization problem. Therefore, our `APPO` is the first offline PbRL algorithm to achieve both provable statistical efficiency and computational efficiency.

---

[2]Our analysis remains valid under an alternative definition based on the reward model error ratio, similar to that used by Zhan et al. (2024a).

[3]Zhan et al. (2024a) consider reward functions defined over trajectories, thus their reward class $\mathcal{G}_r$ is comparable to our $\mathcal{R}^H$. They use bracketing numbers in their bound, but we write here $|\mathcal{P}|$ for simplicity.

**Proof Sketch.** We outline the proof of Theorem 4.1, where the detailed proof is deferred to Appendix D. The key observation is our novel sub-optimality decomposition:

$$V_{1,r^\star}^{\pi^\star} - V_{1,r^\star}^{\pi^t}$$
$$= \underbrace{V_{1,r^\star - \hat{r}}^{\pi^\star} - V_{1,r^\star - \hat{r}}^{\mu}}_{\text{(I) : MLE error}} + \underbrace{V_{1,\hat{r} - r^t}^{\pi^\star} - V_{1,\hat{r} - r^t}^{\mu} - V_{1,r^t}^{\pi^t} + V_{1,r^t}^{\mu} + V_{1,r^t}^{\pi^t} - V_{1,r^t}^{\mu}}_{\text{(II) : Optimization error}} + \underbrace{V_{1,r^t}^{\pi^\star} - V_{1,r^t}^{\pi^t}}_{\text{(III) : Policy update regret}} ,$$

where $r^t = r_{P^\star, f^t}^{\pi^t}$, and the initial state $s_1$ is omitted here for readability. The term (I) is bounded by a standard MLE guarantee (Lemma E.2), while the policy update rule ensures that the sum of terms (III) over $T$ steps is bounded (Lemma D.3). For (II), Assumption 4 and $\lambda > C_{\text{TR}}$ imply that

$$V_{1,\hat{r} - r^t}^{\pi^\star} - V_{1,\hat{r} - r^t}^{\mu} = \mathbb{E}_{\tau^0 \sim \pi^\star, \tau^1 \sim \mu} \left[ r^t(\tau^0) - \hat{r}(\tau^0) - r^t(\tau^1) + \hat{r}(\tau^1) \right]$$
$$\leq C_{\text{TR}} \mathbb{E}_{\tau^0 \sim \pi^\star, \tau^1 \sim \mu} \left[ \left| r^t(\tau^0) - \hat{r}(\tau^0) - r^t(\tau^1) + \hat{r}(\tau^1) \right| \right] \leq \lambda \mathcal{E}(f^t; P^\star, \hat{r}).$$

Observe that APPO approximately solves the optimization problem in (5) (Lemma D.1), and this optimization problem is equivalent to $\arg\min_{f \in \mathcal{F}^H} \{ V_{1,r_{P^\star,f}^{\pi^t}}^{\pi^t} - V_{1,r_{P^\star,f}^{\pi^t}}^{\mu} + \lambda \mathcal{E}(f; P^\star, \hat{r}) \}$. Since $r_{P^\star,f^t}^{\pi^t} = r^t$ and $r_{P^\star,Q^{\pi^t}}^{\pi^t} = r^\star$, it follows that

$$V_{1,r^t}^{\pi^t} - V_{1,r^t}^{\mu} + \lambda \mathcal{E}(f^t; P^\star, \hat{r}) \leq V_{1,r^\star}^{\pi^t} - V_{1,r^\star}^{\mu} + \lambda \mathcal{E}(Q^{\pi^t}; P^\star, \hat{r}) + \epsilon.$$

where $\epsilon$ represents an approximation error. Thus, we obtain (II)$\leq \epsilon$. Combining the results with $V_{1,r^\star}^{\pi^\star} - V_{1,r^\star}^{\hat{\pi}} = \frac{1}{T} \sum_{t=1}^{T} \left( V_{1,r^\star}^{\pi^\star} - V_{1,r^\star}^{\pi^t} \right)$, we complete the proof.

## 5  PRACTICAL IMPLEMENTATION OF APPO

While providing strong statistical guarantees, APPO allows practical implementation with neural networks, leveraging advanced training techniques from deep learning. In this section, we present a practical version of APPO tailored for deep PbRL. The pseudo-code is outlined in Algorithm 4. For practical implementation, we assume the standard discounted MDP setting in deep PbRL (Christiano et al., 2017), where trajectory segments of length $L$ are given and preference labels are assigned to segment pairs.

**Reward Learning.** While our theoretical analysis is based on the maximum likelihood estimator, any reward learning strategy can be employed. This flexibility allows APPO to benefit from state-of-the-arts preference learning methods, such as data augmentation (Park et al., 2022) and active query techniques (Shin et al., 2023; Hwang et al., 2024; Choi et al., 2024).

**Training Value Functions.** Given a parameterized policy $\pi_\theta$ and an action-value function $Q_\phi$, the optimization objective in (5) can be adapted to the discounted setting as follows:

$$\arg\min_\phi \mathbb{E}_{(s,a) \sim d^\mu} \left[ (Q_\phi \circ \pi_\theta)(s) - Q_\phi(s,a) \right] + \lambda \mathbb{E}_{(\tau_0, \tau_1) \sim \mu} \left[ \left| (r_\phi^\theta - \hat{r})(\tau^0) - (r_\phi^\theta - \hat{r})(\tau^1) \right| \right]$$

where $r_\phi^\theta(\tau) = \sum_{l=1}^{L} (Q_\phi(s_l, a_l) - \gamma(Q_\phi \circ \pi_\theta)(s_{l+1}))$ for the segment $\tau = (s_1, a_1, \ldots, s_L, a_L)$. We employ the approximation $P^\star(Q_\phi \circ \pi_\theta)(s_l, a_l) \approx (Q_\phi \circ \pi_\theta)(s_{l+1})$ to avoid the need for a transition model. Additionally, to stabilize training, we apply the clipped double Q-learning trick (Fujimoto et al., 2018; Haarnoja et al., 2018) and maintain a separate value-function $V_\psi$. Given mini-batches of trajectory pairs $\mathcal{B}_{\text{traj}}$ and transition tuples $\mathcal{B}_{\text{tup}}$, each action-value function $Q_{\phi^i}$ is trained by minimizing $\mathcal{L}_{\phi^i}^\lambda = \lambda \mathcal{L}_{\phi^i}^{\text{adv}} + \mathcal{E}_{\phi^i}$ (where $\lambda$ is moved to the first term, without loss of generality), defined as follows:

$$\mathcal{L}_{\phi^i}^{\text{adv}}(\mathcal{B}_{\text{tup}}) = \mathbb{E}_{(s,a) \sim \mathcal{B}_{\text{tup}}} \left[ Q_{\phi^i}(s, \pi_\theta(s)) - Q_{\phi^i}(s,a) \right],$$

$$\text{and } \mathcal{E}_{\phi^i}(\mathcal{B}_{\text{traj}}) = \mathbb{E}_{(\tau^0, \tau^1) \sim \mathcal{B}_{\text{traj}}} \left[ \left| \{ r_{\phi^i}^\psi(\tau^0) - r_{\phi^i}^\psi(\tau^1) \} - \{ \hat{r}(\tau^0) - \hat{r}(\tau^1) \} \right| \right]. \quad (6)$$

Here, we use the notation $r_{\phi^i}^\psi(\tau) = \sum_{l=1}^{L} (Q_{\phi^i}(s_l, a_l) - \gamma V_\psi(s_{h+l}))$, and $\pi_\theta(s)$ denotes an action sampled from $\pi_\theta(\cdot \mid s)$. Given target Q-networks $\{\bar{\phi}^i\}_{i \in \{1,2\}}$, we train $V_\psi$ by minimizing

$$\mathcal{L}_\psi(\mathcal{B}_{\text{tup}}) = \mathbb{E}_{s \sim \mathcal{B}_{\text{tup}}} \left[ \left( V_\psi(s) - \min_{i \in \{1,2\}} Q_{\bar{\phi}^i}(s_{h+1}, \pi_\theta(s_{h+1})) \right)^2 \right], \quad (7)$$

| Dataset & # of feedback | Oracle | MR | PT | DPPO | IPL | APPO (ours) |
|---|---|---|---|---|---|---|
| BPT-500 | 88.33 $\pm4.76$ | 10.08 $\pm7.57$ | 22.87 $\pm9.06$ | 3.93 $\pm4.34$ | 34.73 $\pm13.9$ | **53.52** $\pm13.9$ |
| box-close-500 | 93.40 $\pm3.10$ | **29.12** $\pm13.2$ | 0.33 $\pm1.16$ | 10.20 $\pm11.5$ | 5.93 $\pm5.81$ | 18.24 $\pm15.6$ |
| dial-turn-500 | 75.40 $\pm5.47$ | 61.44 $\pm6.08$ | **68.67** $\pm12.4$ | 26.67 $\pm22.2$ | 31.53 $\pm12.5$ | **80.96** $\pm4.49$ |
| sweep-500 | 98.33 $\pm1.87$ | **86.96** $\pm6.93$ | 43.07 $\pm24.6$ | 10.47 $\pm15.8$ | 27.20 $\pm23.8$ | 26.80 $\pm5.32$ |
| BPT-wall-500 | 56.27 $\pm6.32$ | 0.32 $\pm0.30$ | 0.87 $\pm1.43$ | 0.80 $\pm1.51$ | 8.93 $\pm9.84$ | **64.32** $\pm21.0$ |
| sweep-into-500 | 78.80 $\pm7.96$ | 28.40 $\pm5.47$ | 20.53 $\pm8.26$ | 23.07 $\pm7.02$ | **32.20** $\pm7.35$ | 24.08 $\pm5.91$ |
| drawer-open-500 | 100.00 $\pm0.00$ | **98.00** $\pm2.32$ | 88.73 $\pm11.6$ | 35.93 $\pm11.2$ | 19.00 $\pm13.6$ | 87.68 $\pm10.0$ |
| lever-pull-500 | 98.47 $\pm1.77$ | 79.28 $\pm2.95$ | **82.40** $\pm22.7$ | 10.13 $\pm12.2$ | 31.20 $\pm15.8$ | 75.76 $\pm7.17$ |
| BPT-1000 | 88.33 $\pm4.76$ | 8.48 $\pm5.80$ | 18.27 $\pm10.6$ | 3.20 $\pm3.04$ | 36.67 $\pm17.4$ | **59.04** $\pm19.0$ |
| box-close-1000 | 93.40 $\pm3.10$ | 27.04 $\pm14.5$ | 2.27 $\pm2.86$ | 9.33 $\pm9.60$ | 6.73 $\pm8.41$ | **34.24** $\pm18.5$ |
| dial-turn-1000 | 75.40 $\pm5.47$ | 69.44 $\pm4.70$ | 68.80 $\pm5.50$ | 36.40 $\pm21.9$ | 43.93 $\pm13.4$ | **81.44** $\pm6.73$ |
| sweep-1000 | 98.33 $\pm1.87$ | **87.52** $\pm7.87$ | 29.13 $\pm14.6$ | 8.73 $\pm16.4$ | 38.33 $\pm24.9$ | 17.36 $\pm12.4$ |
| BPT-wall-1000 | 56.27 $\pm6.32$ | 0.48 $\pm0.47$ | 2.13 $\pm2.96$ | 0.27 $\pm0.85$ | 14.07 $\pm11.5$ | **62.96** $\pm18.4$ |
| sweep-into-1000 | 78.80 $\pm7.96$ | 26.00 $\pm5.53$ | 20.27 $\pm7.84$ | **23.33** $\pm7.80$ | 30.40 $\pm7.74$ | 18.16 $\pm11.1$ |
| drawer-open-1000 | 100.00 $\pm0.00$ | 98.40 $\pm2.82$ | 95.40 $\pm7.27$ | 36.47 $\pm7.30$ | 28.53 $\pm18.4$ | **98.56** $\pm2.68$ |
| lever-pull-1000 | 98.47 $\pm1.77$ | **88.96** $\pm3.94$ | 72.93 $\pm10.2$ | 8.53 $\pm9.96$ | 40.40 $\pm17.4$ | 76.96 $\pm4.40$ |
| Average Rank | - | 2.316 | 3.125 | 4.375 | 3.063 | **2.125** |

Table 1: Success rates on Meta-World `medium-replay` dataset with 500 and 1000 preference feedback samples, averaged over 5 random seeds. The results of baselines, Oracle, PT, DPPO, and IPL, are taken from Choi et al. (2024), where Oracle refers to the policy trained using IQL with ground-truth rewards. The abbreviation BPT stands for button-press-topdown.

Intuitively, the term $\mathcal{L}_\phi^{\mathrm{adv}}$ ensures conservatism by regularizing $Q_\phi$ to have lower values near $d^{\pi_\theta}$ and higher values near $d^\mu$. Additional insight can be gained by rearranging the integrand of $\mathcal{E}_\phi$:

$$r_{\phi^i}^\psi(\tau) - \hat{r}(\tau) = \sum_{l=1}^L \left( Q_{\phi^i}(s_l, a_l) - \hat{r}(s_l, a_l) - \gamma V_\psi(s_{l+1}) \right).$$

This expression represents the sum of the TD errors evaluated over the segment $\tau$. Thus, the loss $\mathcal{E}_\phi$ minimizes the difference in trajectory TD errors between $\tau^0$ and $\tau^1$.

**Training Policy.** We directly optimize the policy using the loss function in (8). The entropy regularization term is similar to that in SAC (Haarnoja et al., 2018), except that we use a randomly sampled $Q_{\phi^i}$ instead of the clipped value $\min_{i \in [1,2]} Q_{\phi^i}$. The policy loss is given by:

$$\mathcal{L}_\theta(\mathcal{B}_{\mathrm{tup}}) = \mathbb{E}_{s \sim \mathcal{B}_{\mathrm{tup}}} \left[ Q_{\phi^i}(s, \pi_\theta(s)) - \alpha \pi_\theta(s, \pi_\theta(s)) \right], \ i \sim \mathrm{Unif}\{1, 2\} \tag{8}$$

## 6 EXPERIMENTS

**Datasets and Evaluation.** We evaluate our proposed algorithm on the Meta-World (Yu et al., 2020) `medium-replay` and `medium-expert` datasets from Choi et al. (2024). Our main experiments use the `medium-replay` dataset, while the experiments with the `medium-expert` dataset are presented in Appendix F. These datasets have a favorable property: they are not learnable with incorrect rewards (random or constant). This property is crucial for evaluating offline RL algorithms since their survival instinct can allow them to perform well even with completely incorrect reward signals (Li et al., 2024). For more details on the dataset, see Choi et al. (2024). Following the experiment protocol of Choi et al. (2024), the preference dataset consists of pairs of randomly sampled trajectory segments of length 25. The preference label is generated based on the ground truth reward, where a $(0, 1)$ label is assigned if the trajectory rewards differ by more than a threshold of 12.5, and a $(0.5, 0.5)$ label is assigned otherwise. We evaluate algorithm performance using the success rate for each task, which indicates whether the agent successfully completes the task.

**Algorithms.** We consider four offline PbRL algorithms as baselines: Markovian Reward (MR), Preference Transformer (PT) (Kim et al., 2023), Direct Preference-based Policy Optimization

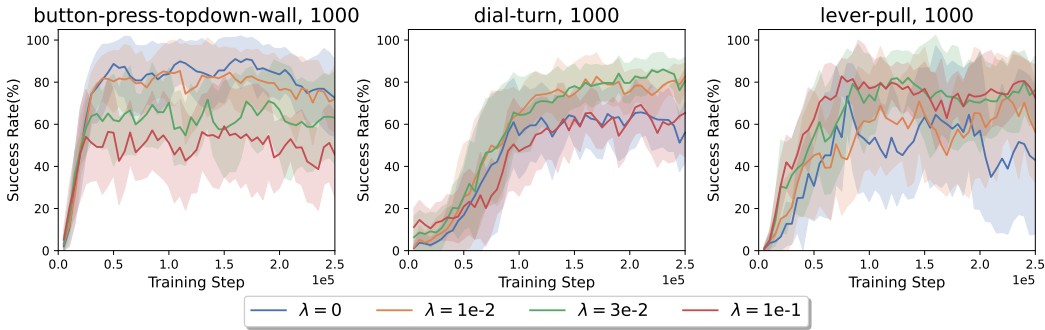

Figure 1: Effect of the conservatism regularizer $\lambda$.

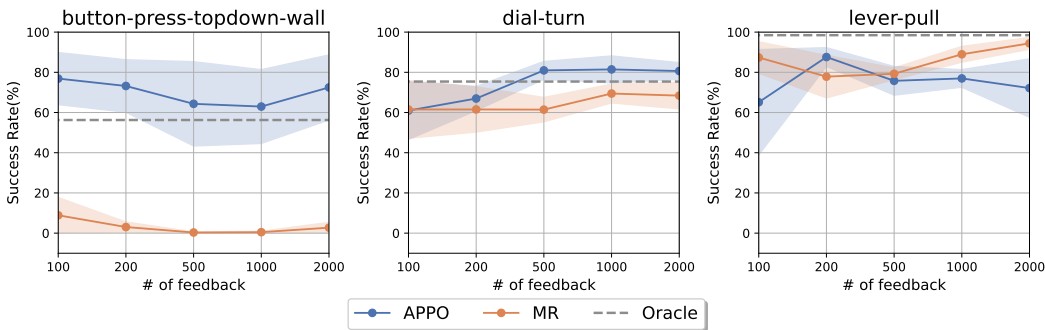

Figure 2: Success rates of APPO and MR, with varying number of preference feedback samples.

(DPPO) (An et al., 2023), and Inverse Preference Learning (IPL) (Hejna & Sadigh, 2024). MR is an instance of IQL (Kostrikov et al., 2022) trained with a Markovian reward model, while PT assumes a general sequential reward model implemented using a Transformer (Vaswani, 2017) architecture. DPPO directly optimizes the policy without using a reward model, while the other baseline methods are based on IQL (Kostrikov et al., 2022). We evaluate the practical version of APPO in Algorithm 4, using the same reward model as MR and setting $\lambda = 0.03$. Further details are in Appendix G.

## 6.1 EVALUATION RESULTS

Table 1 shows the performance of algorithms on the Meta-World control tasks. APPO outperforms or shows comparable performance in almost every dataset. Notably, APPO outperforms the policy trained with ground truth rewards in the dial-turn and button-press-topdown-wall datasets. We also observe that MR is a strong baseline, as reported in previous works (Hejna & Sadigh, 2024; Choi et al., 2024). These results suggest that APPO performs comparably to state-of-the-art baselines, even with a provable statistical guarantee.

**Effect of Conservatism Regularizer.** We investigate the effect of conservatism regularizer $\lambda$, the coefficient to balance the adversarial loss $\mathcal{L}_\phi^{\mathrm{adv}}$ and the trajectory-pair $\ell_1$ loss $\mathcal{E}_\phi$. In Figure 1, APPO learns successfully with a wide range of $\lambda$, but a properly tuned $\lambda$ improves performance and stability. We note that APPO has only one algorithmic hyperparameter $\lambda$, whereas IQL-based algorithms (MR, PT, IPL) require at least two (expectile parameter and temperature). DPPO, on the other hand, has two hyperparameters (conservatism regularizer and smoothness regularizer).

**Effect of Preference Dataset Size.** In PbRL, learning from small preference datasets is desired for cost-efficient learning. We evaluate the effect of preference dataset size on APPO's performance, varying the number of feedback samples from 100 to 2000. Figure 2 shows that APPO is robust to preference dataset size, displaying variance comparable to MR, a strong baseline, as shown in Table 1. Note that APPO outperforms a policy trained with ground truth rewards, using only 100 preference feedback samples.

## REPRODUCIBILITY

We describe the experimental details in Section 6 and Section G, including training protocol and neural network architecture. Our code is available at `https://github.com/oh-lab/APPO.git`. As explained in Section 6, we use the Meta-World `medium-replay` dataset from Choi et al. (2024), which is available in their official repository along with download instructions.

## ACKNOWLEDGEMENTS

This work was supported by the National Research Foundation of Korea (NRF) grant funded by the Korea government (MSIT) (No. RS-2022-NR071853 and RS-2023-00222663) and by AI-Bio Research Grant through Seoul National University.

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

## A   ADDITIONAL RELATED WORK

**Empirical PbRL.** Incorporating preference feedback into reinforcement learning has been explored through various approaches. One standard method involves training a reward model from preferences, which is then used to train a standard RL algorithm (Christiano et al., 2017; Ibarz et al., 2018). A variety of techniques have emerged in this area, including unsupervised pre-training (Lee et al., 2021), exploration driven by uncertainty (Liang et al., 2022), data augmentation (Park et al., 2022), and meta-learning (Hejna III & Sadigh, 2023), to list a few. Another prominent line of research focuses on preference learning via active query methods (Shin et al., 2023; Hwang et al., 2024; Choi et al., 2024), which have shown strong empirical performance.

Beyond the conventional Markov reward model, some studies have proposed alternative reward structures. For example, Kim et al. (2023) employed transformer architectures for reward modeling, while Liu et al. (2022) and Hejna & Sadigh (2024) explored learning action-value functions rather than directly modeling rewards. Several approaches bypass explicit reward models altogether, instead optimizing policies directly (An et al., 2023; Kang et al., 2023; Hejna et al., 2024).

## B   DETAILS ON THE POLICY EVALUATION SUBROUTINE

We present a simple policy evaluation subroutine in Algorithm 3. It requires online rollouts and access to the reference policy. The idea of policy evaluation via online rollouts is adopted from Chang et al. (2024), while the analysis follows standard methods.

---

**Algorithm 3** PE: Monte Carlo Policy Evaluation

---

1: **Input:** Reference policy $\mu$, Current policy $\pi^t$, Estimated reward $\hat{r}$, Number of rollout $K$
2: **for** $h \in [H]$ **do**
3:     Collect $K$ i.i.d. trajectories $\{(s_1^k, a_1^k, \ldots, s_H^k, a_H^k)\}_{k=1}^K$
4:     where $a_j^k \sim \mu_j(\cdot \mid s_j^k)$ for $j < h$, $a_h^k \sim \frac{1}{2}(\mu_h + \pi_h^t)(\cdot \mid s_h^k)$, and $a_j^k \sim \pi_j^t(\cdot \mid s_j^k)$ for $j > h$
5:     Compute $q_h^k = \sum_{j=h}^H \hat{r}(s_j^k, a_j^k)$, then set $\mathcal{D}_h^t = \{(s_h^k, a_h^k, q_h^k)\}_{k=1}^K$
6:     Least square value function estimation $\bar{Q}_h^t = \arg\min_{f \in \mathcal{F}} \frac{1}{K} \sum_{(s,a,q) \in \mathcal{D}_h^t} (f(s,a) - q)^2$
7: **end for**
8: Return $\{\bar{Q}_h^t\}_{h=1}^H$

---

We have the following guarantee.

**Lemma B.1.** *With probability at least $1 - \delta$, Algorithm 3 guarantees that, for every $(t, h) \in [T] \times [H]$,*

$$\mathbb{E}_{s \sim d_h^\mu, a \sim \frac{1}{2}(\mu^h + \pi_h^t)} \left[ \left( \bar{Q}_h^t(s,a) - Q_{h,r^t}^{\pi^t}(s,a) \right)^2 \right] \leq \frac{c_3 R^2 \log(TH|\mathcal{F}|/\delta)}{K_2} =: \epsilon_{PE}^2$$

*where $c_3$ is an absolute constant.*

*Proof.* Since $\left\| Q_{h,r}^\pi \right\|_\infty \leq R$ for any policy $\pi$ and $r \in \mathcal{R}^H$, Lemma E.4 with $B = R$ and $K = K_2$ leads to

$$\mathbb{E}_{s \sim d_h^\mu, a \sim \frac{1}{2}(\mu^h + \pi_h^t)} \left[ \left( \bar{Q}_h^t(s,a) - Q_{h,r^t}^{\pi^t}(s,a) \right)^2 \right] \leq \frac{c_3 R^2 \log(|\mathcal{F}|/\delta)}{K_2}$$

for any fixed $(t,h) \in [T] \times [H]$. The union bound over all $(t,h) \in [T] \times [H]$ concludes the proof.  $\square$

## C    THEORETICAL ANALYSIS OF APPO-ROLLOUT

In this section, we provide theoretical analyses of APPO-rollout, a naïve algorithm for solving the optimization problem (2). The ideas presented in this section are relevant to the proof of Theorem 4.1, and the results are valuable for comparison with related works.

Before stating the theorem, we define step-wise concentrability, which is always bounded by $C_{\text{TR}}$.

**Definition 2** (Step-wise concentrability). $C_{ST} = \max_{h \in [H]} \sup_{(s,a) \in \mathcal{S} \times \mathcal{A}} \frac{d_h^{\pi^\star}(s,a)}{d_h^\mu(s,a)}$

**Lemma C.1.** *It always holds that $C_{ST} \leq C_{TR}$.*

*Proof.* For a fixed pair $(s, a)$, consider the set of trajectories $\mathcal{T}(s, a) := \{\tau = (s_1, a_1, \dots, s_H, a_H) : s_h = s, a_h = a\}$. Then we have that

$$d_h^\pi(s, a) = \int_{\mathcal{T}(s,a)} d^\pi(\tau) d\tau.$$

for any fixed policy $\pi$. Therefore, for every $(s, a) \in \mathcal{S} \times \mathcal{A}$, we have that

$$\frac{d_h^{\pi^\star}(s, a)}{d_h^\mu(s, a)} = \frac{\int_{\mathcal{T}(s,a)} d^{\pi^\star}(\tau) d\tau}{\int_{\mathcal{T}(s,a)} d^\mu(\tau) d\tau} \leq \sup_\tau \frac{d^{\pi^\star}(\tau)}{d^\mu(\tau)} = C_{\text{TR}}.$$

Taking the supremum on both sides concludes the proof.    □

**Theorem C.2.** *Suppose Assumptions 1 and 4 hold. With probability at least $1 - \delta$, Algorithm 1 with $\lambda = \Theta(C_{TR}), \lambda > C_{TR}, \eta = \sqrt{\frac{2 \log |\mathcal{A}|}{R^2 T}}$ achieves*

$$V_{1,r^\star}^{\pi^\star} - V_{1,r^\star}^{\hat\pi}$$
$$\leq \mathcal{O}\left( \sqrt{\log \frac{|\mathcal{R}|}{\delta}} \left( \frac{C_{TR} \kappa \sqrt{H}}{\sqrt{M}} + \frac{R}{\sqrt{K_1}} + \frac{R}{\sqrt{N}} \right) + RH \sqrt{\frac{\log |\mathcal{A}|}{T}} + RH \sqrt{\frac{C_{ST}}{K_2} \log \frac{TH|\mathcal{F}|}{\delta}} \right).$$

*Setting $T = \Theta\left( \frac{R^2 H^2 \log |\mathcal{A}|}{\epsilon^2} \right), N = K_1 = \Theta\left( \frac{R^2 \log(|\mathcal{R}|/\delta)}{\epsilon^2} \right), M = \Theta\left( \frac{C_{TR}^2 \kappa^2 H \log(|\mathcal{R}|/\delta)}{\epsilon^2} \right)$, and $K_2 = \Theta\left( \frac{R^2 H^2 C_{ST} \log(TH|\mathcal{F}|/\delta)}{\epsilon^2} \right)$, Algorithm 2 achieves $\epsilon$-optimal policy, i.e. $V_{1,r^\star}^{\pi^\star} - V_{1,r^\star}^{\bar\pi} \leq \epsilon$.*

**Discussion on Theorem C.2.** We compare this bound with PbRL algorithms that assume a known transition model or allow online rollouts. In comparison to FREEHAND (Zhan et al., 2024a), APPO-rollout achieves a nearly identical rate for labeled data, but unlike FREEHAND, it requires additional unlabeled trajectories. This represents a trade-off between statistical efficiency and computational complexity, as FREEHAND relies on solving a nearly intractable nested optimization problem. Another comparable algorithm is DR-PO (Chang et al., 2024), which establishes a sample complexity of $\Theta\left( \frac{(C_{\text{TR}} + C_{\text{SFT}}) \kappa^2 \log(|\mathcal{R}|/\delta)}{\epsilon^2} \right)$ for labeled data. Unlike APPO-rollout, DR-PO assumes homogeneous rewards, which removes dependence on $H$ from the bound. While their bound is tighter in $C_{\text{TR}}$, it comes at the cost of dependence on an additional factor, $C_{\text{SFT}} = \sup_{\pi \in D} \sup_\tau \frac{d^\pi(\tau)}{d^\mu(\tau)}$, where $D$ is a set of policies close to $\mu$ in terms of KL divergence. This additional term arises because DR-PO does not explicitly ensure conservatism.

For simplicity, we introduce some notation regarding optimization objectives in Algorithm 1. For $r, \tilde{r} \in \mathcal{R}^H$, we define

$$\hat{\mathcal{L}}_{\text{opt}}^t(r; \tilde{r}) := \mathbb{E}_{\tau \sim \mathcal{D}_{\text{rollout}}}[r(\tau)] - \mathbb{E}_{\tau \sim \mathcal{D}_{\text{traj}}}[r(\tau)] + \lambda \hat{\mathcal{E}}_{\mathcal{D}_{\text{traj}}}(r; \tilde{r})$$

and its population version as

$$\mathcal{L}_{\text{opt}}^t(r; \tilde{r}) := \mathbb{E}_{\tau \sim \pi^t}[r(\tau)] - \mathbb{E}_{\tau \sim \mu}[r(\tau)] + \lambda \mathcal{E}(r; \tilde{r}).$$

## C.1 OPTIMIZATION ERROR

In this section, we prove that the (finite-sample) optimization objective $\hat{\mathcal{L}}^t_{\text{opt}}(r; \hat{r})$ is close to its population version, $\mathcal{L}^t_{\text{opt}}(r; \tilde{r})$. The result ensures that $r^t$ is a good approximation of the solution to the optimization program with infinite samples, i.e.

$$r^t \approx \arg\min_{r \in \mathcal{R}^H} \mathcal{L}^t_{\text{opt}}(r; \hat{r}).$$

**Lemma C.3.** *With probability at least $1 - \delta/2$, for all $t \in [T]$, we have*

$$\mathcal{L}_{opt}(r^t; \hat{r}) \leq \mathcal{L}_{opt}(r^\star; \hat{r}) + 2\tilde{\epsilon}_{approx}$$

*where $\tilde{\epsilon}_{approx}$ is defined in Lemma C.4.*

*Proof.* We have the following decomposition:

$$\mathcal{L}^t_{\text{opt}}(r^t; \hat{r}) - \mathcal{L}^t_{\text{opt}}(r^\star; \hat{r})$$
$$= \underbrace{\mathcal{L}^t_{\text{opt}}(r^t; \pi^t) - \hat{\mathcal{L}}^t_{\text{opt}}(r^t; \hat{r})}_{\text{(I)}} + \underbrace{\hat{\mathcal{L}}^t_{\text{opt}}(r^t; \hat{r}) - \hat{\mathcal{L}}^t_{\text{opt}}(r^\star; \hat{r})}_{\text{(II)}} + \underbrace{\hat{\mathcal{L}}^t_{\text{opt}}(r^\star; \hat{r}) - \mathcal{L}^t_{\text{opt}}(r^\star; \hat{r})}_{\text{(III)}}$$

Conditioned on the event defined by Lemma D.2, (I) and (III) are bounded by $\epsilon_{opt}$. Moreover, the optimality of $r^t$ implies (II)$\leq 0$. □

**Lemma C.4.** *With probability at least $1 - \delta/2$, for every $t \in [T]$ and $r \in \mathcal{R}^H$, it holds that*

$$\left| \mathcal{L}^t_{opt}(r; \hat{r}) - \hat{\mathcal{L}}^t_{opt}(r; \hat{r}) \right| \leq R\sqrt{\frac{\log(6|\mathcal{R}|/\delta)}{2K_1}} + 2R\sqrt{\frac{2\log(6|\mathcal{R}|/\delta)}{N}} := \tilde{\epsilon}_{approx}$$

*Proof.* Fix $r \in \mathcal{R}^H$, and note that

$$\left| \mathcal{L}^t_{\text{opt}}(r; \hat{r}) - \hat{\mathcal{L}}^t_{\text{opt}}(r; \hat{r}) \right|$$
$$\leq \left| \mathbb{E}_{\tau \sim \mathcal{D}^t_{\text{rollout}}}[r(\tau)] - \mathbb{E}_{\tau \sim \pi^t}[r(\tau)] \right| + \left| \mathbb{E}_{\tau \sim \mathcal{D}_{\text{traj}}}[r(\tau)] - \mathbb{E}_{\tau \sim \mu}[r(\tau)] \right|$$
$$+ \left| \mathbb{E}_{(\tau^0, \tau^1) \sim \mathcal{D}_{\text{traj}}} \left[ (r - \hat{r})(\tau^0) - (r - \hat{r})(\tau^1) \right] - \mathbb{E}_{(\tau^0, \tau^1) \sim \mu} \left[ (r - \hat{r})(\tau^0) - (r - \hat{r})(\tau^1) \right] \right|.$$

Since $|r(\tau)| \leq R$ and $|(r - \hat{r})(\tau)| \leq R$ for any trajectory $\tau$, each term can be bounded by Hoeffding inequality. Specifically, each of these three events occurs with probability at least $1 - \delta/6$:

$$\left| \mathbb{E}_{\tau \sim \mathcal{D}^t_{\text{rollout}}}[r(\tau)] - \mathbb{E}_{\tau \sim \pi^t}[r(\tau)] \right| \leq R\sqrt{\frac{\log(6/\delta)}{2K_1}},$$

$$\left| \mathbb{E}_{\tau \sim \mathcal{D}_{\text{traj}}}[r(\tau)] - \mathbb{E}_{\tau \sim \mu}[r(\tau)] \right| \leq R\sqrt{\frac{\log(6/\delta)}{2N}},$$

$$\left| \mathbb{E}_{(\tau^0, \tau^1) \sim \mathcal{D}_{\text{traj}}} \left[ (r - \hat{r})(\tau^0) - (r - \hat{r})(\tau^1) \right] - \mathbb{E}_{(\tau^0, \tau^1) \sim \mu} \left[ (r - \hat{r})(\tau^0) - (r - \hat{r})(\tau^1) \right] \right| \leq 2R\sqrt{\frac{\log(6/\delta)}{2N}}.$$

Taking union bound over these events and all $r \in \mathcal{R}^H$, with probability at least $1 - \delta/2$, it holds that

$$\left| \mathcal{L}^t_{\text{opt}}(r; \hat{r}) - \hat{\mathcal{L}}^t_{\text{opt}}(r; \hat{r}) \right| \leq R\sqrt{\frac{\log(6|\mathcal{R}|/\delta)}{2K_1}} + R\sqrt{\frac{\log(6|\mathcal{R}|/\delta)}{2N}} + 2R\sqrt{\frac{\log(6|\mathcal{R}|/\delta)}{2N}}$$

$$\leq R\sqrt{\frac{\log(6|\mathcal{R}|/\delta)}{2K_1}} + 2R\sqrt{\frac{2\log(6|\mathcal{R}|/\delta)}{N}}$$

for every $r \in \mathcal{R}^H$. □

## C.2  POLICY UPDATE

We present the guarantee regarding the policy update steps. The proofs in this section are based on the standard analysis of the natural policy gradient (also referred to as trust region policy optimization) (Cai et al., 2020; Chang et al., 2024).

**Lemma C.5.** *With probability at least $1 - \delta/4$, it holds that*

$$\frac{1}{T}\sum_{t=1}^{T}\left(V_{1,r^t}^{\pi^\star}(s_1) - V_{1,r^t}^{\pi^t}(s_1)\right) \leq RH\sqrt{\frac{\log|\mathcal{A}|}{2T}} + 2H\epsilon_{PE}\sqrt{2C_{ST}}$$

*Proof of Lemma C.5.* The performance difference lemma (Lemma E.1) implies that

$$\sum_{t=1}^{T}\left(V_{1,r^t}^{\pi^\star}(s_1) - V_{1,r^t}^{\pi^t}(s_1)\right)$$

$$= \sum_{t=1}^{T}\mathbb{E}_{\pi^\star}\left[\sum_{h=1}^{H}\langle Q_{h,r^t}^{\pi^t}(s_h,\cdot), \pi_h^\star(\cdot \mid s_h) - \pi_h^t(\cdot \mid s_h)\rangle\right]$$

$$= \underbrace{\sum_{t=1}^{T}\sum_{h=1}^{H}\mathbb{E}_{s\sim d_h^{\pi^\star}}\left[\langle\bar{Q}_h^t(s,\cdot), \pi_h^\star(\cdot \mid s) - \pi_h^t(\cdot \mid s)\rangle\right]}_{(I)}$$

$$+ \underbrace{\sum_{t=1}^{T}\sum_{h=1}^{H}\mathbb{E}_{s\sim d_h^{\pi^\star}}\left[\langle(Q_{h,r^t}^{\pi^t} - \bar{Q}_h^t)(s,\cdot), \pi_h^\star(\cdot \mid s_h) - \pi_h^t(\cdot \mid s_h)\rangle\right]}_{(II)}$$

**Bounding (I).**  Decompose the inner product inside the expectation:

$$\langle\eta\bar{Q}_h^t(s_h,\cdot), \pi_h^\star(\cdot \mid s) - \pi_h^t(\cdot \mid s)\rangle$$

$$\langle\eta\bar{Q}_h^t(s_h,\cdot), \pi_h^\star(\cdot \mid s) - \pi_h^{t+1}(\cdot \mid s)\rangle + \langle\eta\bar{Q}_h^t(s_h,\cdot), \pi_h^{t+1}(\cdot \mid s) - \pi_h^t(\cdot \mid s)\rangle$$

$$\leq \langle\eta\bar{Q}_h^t(s_h,\cdot), \pi_h^\star(\cdot \mid s) - \pi_h^{t+1}(\cdot \mid s)\rangle + \eta\left\|\bar{Q}_h^t(s_h,\cdot)\right\|_\infty \left\|\pi_h^\star(\cdot \mid s) - \pi_h^{t+1}(\cdot \mid s)\right\|_1$$

$$\leq \langle\eta\bar{Q}_h^t(s_h,\cdot), \pi_h^\star(\cdot \mid s) - \pi_h^{t+1}(\cdot \mid s)\rangle + \eta R\left\|\pi_h^\star(\cdot \mid s) - \pi_h^{t+1}(\cdot \mid s)\right\|_1 \tag{9}$$

where we use Hölder's inequality with the fact that $\left\|\bar{Q}_h^t\right\|_\infty \leq R$. Now recall that the policy update step (Line 7) in Algorithm 1 leads to

$$\pi_h^{t+1}(\cdot \mid s) = \frac{1}{Z_h^t(s)}\pi_h^t(\cdot \mid s)\exp\left(\eta\bar{Q}_h^t(s,\cdot)\right)$$

where $Z_h^t(s) = \sum_{a\in\mathcal{A}}\pi_h^t(a \mid s)\exp\left(\eta\bar{Q}_h^t(s,a)\right)$. Using the relationship $\eta\bar{Q}_h^t(s,a) = \log Z_h^t(s) + \log\pi_h^{t+1}(a \mid s) - \log\pi_h^t(a \mid s)$, it holds that

$$\langle\eta\bar{Q}_h^t(s_h,\cdot), \pi_h^\star(\cdot \mid s) - \pi_h^{t+1}(\cdot \mid s)\rangle$$

$$= \langle\log Z_h^t(s) + \log\pi_h^{t+1}(\cdot \mid s) - \log\pi_h^t(\cdot \mid s), \pi_h^\star(\cdot \mid s) - \pi_h^{t+1}(\cdot \mid s)\rangle$$

$$= \langle\log\pi_h^{t+1}(\cdot \mid s) - \log\pi_h^t(\cdot \mid s), \pi^\star(\cdot \mid s) - \pi_h^{t+1}(\cdot \mid s)\rangle$$

$$= \langle\log\pi_h^{t+1}(\cdot \mid s) - \log\pi_h^t(\cdot \mid s), \pi^\star(\cdot \mid s)\rangle - D_{KL}\left(\pi_h^{t+1}(\cdot \mid s)\|\pi_h^t(\cdot \mid s)\right)$$

$$= \langle\log\frac{\pi_h^{t+1}(\cdot \mid s)}{\pi_h^\star(\cdot \mid s)} + \log\frac{\pi_h^\star(\cdot \mid s)}{\pi_h^t(\cdot \mid s)}, \pi_h^\star(\cdot \mid s)\rangle - D_{KL}\left(\pi_h^{t+1}(\cdot \mid s)\|\pi_h^t(\cdot \mid s)\right)$$

$$= D_{KL}\left(\pi_h^\star(\cdot \mid s)\|\pi_h^t(\cdot \mid s)\right) - D_{KL}\left(\pi_h^\star(\cdot \mid s)\|\pi_h^{t+1}(\cdot \mid s)\right) - D_{KL}\left(\pi_h^{t+1}(\cdot \mid s)\|\pi_h^t(\cdot \mid s)\right)$$

$$\leq D_{KL}\left(\pi_h^\star(\cdot \mid s)\|\pi_h^t(\cdot \mid s)\right) - D_{KL}\left(\pi_h^\star(\cdot \mid s)\|\pi_h^{t+1}(\cdot \mid s)\right) - \frac{1}{2}\left\|\pi_h^\star(\cdot \mid s) - \pi_h^{t+1}(\cdot \mid s)\right\|_1^2$$

where the second equality holds since $Z_h^t(s)$ is a constant given $s$, and the last inequality holds due to Pinsker's inequality. Combining this bound with (9), we obtain

$$
\begin{aligned}
&\sum_{t=1}^{T} \langle \eta \bar{Q}_h^t(s_h, \cdot), \pi_h^\star(\cdot \mid s) - \pi_h^t(\cdot \mid s) \rangle \\
&= \sum_{t=1}^{T} \left( D_{KL}\left(\pi_h^\star(\cdot \mid s) \| \pi_h^t(\cdot \mid s)\right) - D_{KL}\left(\pi_h^\star(\cdot \mid s) \| \pi_h^{t+1}(\cdot \mid s)\right) \right) \\
&\quad + \sum_{t=1}^{T} \left( \eta R \left\| \pi_h^\star(\cdot \mid s) - \pi_h^{t+1}(\cdot \mid s) \right\|_1 - \frac{1}{2} \left\| \pi_h^\star(\cdot \mid s) - \pi_h^{t+1}(\cdot \mid s) \right\|_1^2 \right) \\
&\leq \sum_{t=1}^{T} \left( D_{KL}\left(\pi_h^\star(\cdot \mid s) \| \pi_h^t(\cdot \mid s)\right) - D_{KL}\left(\pi_h^\star(\cdot \mid s) \| \pi_h^{t+1}(\cdot \mid s)\right) \right) + \sum_{t=1}^{T} \frac{\eta^2 R^2}{2} \\
&= D_{KL}\left(\pi_h^\star(\cdot \mid s) \| \pi_h^1(\cdot \mid s)\right) - D_{KL}\left(\pi_h^\star(\cdot \mid s) \| \pi_h^{T+1}(\cdot \mid s)\right) + \frac{\eta^2 R^2 T}{2} \\
&\leq \log |\mathcal{A}| + \frac{\eta^2 R^2 T}{2}
\end{aligned}
$$

where the first inequality holds since $\forall x \in \mathbb{R} \, ax - x^2/2 \leq a^2/2$, and the second inequality holds due to the fact that $\pi_h^1 = \mathrm{Unif}(\mathcal{A})$. Finally, setting $\eta = \sqrt{\frac{2 \log |\mathcal{A}|}{R^2 T}}$, (I) is bounded by

$$
\begin{aligned}
\text{(I)} &= \sum_{h=1}^{H} \mathbb{E}_{s \sim d_h^{\pi^\star}} \left[ \sum_{t=1}^{T} \langle \bar{Q}_h^t(s, \cdot), \pi^\star(\cdot \mid s) - \pi^t(\cdot \mid s) \rangle \right] \\
&\leq \sum_{h=1}^{H} \frac{\log |\mathcal{A}|}{\eta} + \frac{\eta R^2 T}{2} = RH \sqrt{\frac{T \log |\mathcal{A}|}{2}}
\end{aligned}
$$

**Bounding (II).** We condition on the event defined by Lemma B.1. Then we have

$$
\begin{aligned}
&\left| \mathbb{E}_{s \sim d_h^{\pi^\star}} \left[ \langle (Q_{h,r^t}^{\pi^t} - \bar{Q}_h^t)(s, \cdot), \pi_h^\star \rangle \right] \right| \\
&= \left| \mathbb{E}_{(s,a) \sim d_h^{\pi^\star}} \left[ Q_{h,r^t}^{\pi^t}(s, a) - \bar{Q}_h^t(s, a) \right] \right| \\
&\leq \sqrt{ \mathbb{E}_{(s,a) \sim d_h^{\pi^\star}} \left[ \left( Q_{h,r^t}^{\pi^t}(s, a) - \bar{Q}_h^t(s, a) \right)^2 \right] } \\
&\leq \sqrt{ 2 \left( \max_{h \in [H]} \sup_{(s,a) \in \mathcal{S} \times \mathcal{A}} \frac{d_h^{\pi^\star}(s, a)}{d_h^\mu(s, a)} \right) \mathbb{E}_{s \sim d_h^\mu, a \sim \frac{1}{2}(\pi_h^t + \mu_h)} \left[ \left( Q_{h,r^t}^{\pi^t}(s, a) - \bar{Q}_h^t(s, a) \right)^2 \right] } \\
&\leq \sqrt{ 2 C_{\mathrm{ST}} \epsilon_{\mathrm{PE}}^2 }
\end{aligned}
$$

where the first inequality holds due to Jensen's inequality, the second inequality uses importance sampling, and the last inequality uses Lemma B.1.

$$\left| \mathbb{E}_{s \sim d_h^{\pi^\star}} \left[ \langle (Q_{h,r^t}^{\pi^t} - \bar{Q}_h^t)(s, \cdot), \pi_h^t \rangle \right] \right|$$

$$= \left| \mathbb{E}_{s \sim d_h^{\pi^\star}, a \sim \pi_h^t} \left[ Q_{h,r^t}^{\pi^t}(s, a) - \bar{Q}_h^t(s, a) \right] \right|$$

$$\leq \sqrt{ \mathbb{E}_{s \sim d_h^{\pi^\star}, a \sim \pi_h^t} \left[ \left( Q_{h,r^t}^{\pi^t}(s, a) - \bar{Q}_h^t(s, a) \right)^2 \right] }$$

$$\leq \sqrt{ 2 \left( \max_{h \in [H]} \sup_{s \in \mathcal{S}} \frac{d_h^{\pi^\star}(s)}{d_h^\mu(s)} \right) \mathbb{E}_{s \sim d_h^\mu, a \sim \frac{1}{2}(\pi_h^t + \mu_h)} \left[ \left( Q_{h,r^t}^{\pi^t}(s, a) - \bar{Q}_h^t(s, a) \right)^2 \right] }$$

$$\leq \sqrt{ 2 \left( \max_{h \in [H]} \sup_{s \in \mathcal{S}} \frac{d_h^{\pi^\star}(s)}{d_h^\mu(s)} \right) \mathbb{E}_{s \sim d_h^\mu, a \sim \frac{1}{2}(\pi_h^t + \mu_h)} \left[ \left( Q_{h,r^t}^{\pi^t}(s, a) - \bar{Q}_h^t(s, a) \right)^2 \right] }$$

$$\leq \sqrt{ 2 C_{\mathrm{ST}} \epsilon_{\mathrm{PE}}^2 }.$$

Therefore, we obtain the bound

$$\text{(II)} \leq \sum_{t=1}^T \sum_{h=1}^H \left| \mathbb{E}_{s \sim d_h^{\pi^\star}} \left[ \langle (Q_{h,r^t}^{\pi^t} - \bar{Q}_h^t)(s, \cdot), \pi_h^\star(\cdot \mid s_h) \rangle \right] \right|$$

$$+ \sum_{t=1}^T \sum_{h=1}^H \left| \mathbb{E}_{s \sim d_h^{\pi^\star}} \left[ \langle (Q_{h,r^t}^{\pi^t} - \bar{Q}_h^t)(s, \cdot), \pi_h^t(\cdot \mid s_h) \rangle \right] \right|$$

$$\leq 2 T H \epsilon_{\mathrm{PE}} \sqrt{2 C_{\mathrm{ST}}}.$$

We conclude the proof by combining the bounds on (I) and (II). $\qquad\square$

Now we prove Theorem C.2 based on the lemmas.

*Proof of Theorem C.2.* We condition on the event defined by Lemma E.2 (with $\delta' = \delta/4$), Lemma C.3, and Lemma C.5, which hold simultaneously with probability at least $1 - \delta$. Consider the following sub-optimality decomposition at step $t$:

$$V_{1,r^\star}^{\pi^\star} - V_{1,r^\star}^{\pi^t} = V_{1,r^\star}^{\pi^\star} - V_{1,\hat{r}}^{\pi^\star} + V_{1,\hat{r}}^{\pi^\star} - V_{1,r^t}^{\pi^\star} + V_{1,r^t}^{\pi^\star} - V_{1,r^\star}^{\pi^t} + V_{1,r^t}^{\pi^t} - V_{1,r^t}^{\pi^t}$$

$$= \underbrace{V_{1,r^\star-\hat{r}}^{\pi^\star} - V_{1,r^\star-\hat{r}}^\mu}_{\text{(I) : MLE estimation error}}$$

$$+ \underbrace{V_{1,\hat{r}-r^t}^{\pi^\star} - V_{1,\hat{r}-r^t}^\mu - V_{1,r^\star}^{\pi^t} + V_{1,r^\star}^\mu + V_{1,r^t}^{\pi^t} - V_{1,r^t}^\mu}_{\text{(II) : Optimization error}}$$

$$+ \underbrace{V_{1,r^t}^{\pi^\star} - V_{1,r^t}^{\pi^t}}_{\text{(III) : Policy update regret}}, \tag{10}$$

where we omit the initial state $s_1$ for simplicity.

**Bounding (I).** Since we condition on the event defined by Lemma E.2, we have

$$\text{(I)} = V_{1,r^\star-\hat{r}}^{\pi^\star} - V_{1,r^\star-\hat{r}}^\mu$$

$$= \mathbb{E}_{\tau^0 \sim \pi^\star, \tau^1 \sim \mu} \left[ r^\star(\tau^0) - r^\star(\tau^1) - \hat{r}(\tau^0) + \hat{r}(\tau^1) \right]$$

$$\leq \sqrt{ \mathbb{E}_{\tau^0 \sim \pi^\star, \tau^1 \sim \mu} \left[ |r^\star(\tau^0) - r^\star(\tau^1) - \hat{r}(\tau^0) + \hat{r}(\tau^1)|^2 \right] }$$

$$\leq \sqrt{ C_{\mathrm{TR}} \mathbb{E}_{\tau^0, \tau^1 \sim \mu} \left[ |r^\star(\tau^0) - r^\star(\tau^1) - \hat{r}(\tau^0) + \hat{r}(\tau^1)|^2 \right] }$$

$$\leq \sqrt{ C_{\mathrm{TR}} } \epsilon_r(\delta/4).$$

**Bounding (II).** We can relate the terms $V_{1,\hat{r}-r^t}^{\pi^\star} - V_{1,\hat{r}-r^t}^{\mu}$ to $\mathcal{E}(r^t; P^\star, \hat{r})$. By Assumption 4, we have that

$$
\begin{aligned}
& V_{1,\hat{r}-r^t}^{\pi^\star} - V_{1,\hat{r}-r^t}^{\mu} \\
& = \mathbb{E}_{\tau^0 \sim \pi^\star, \tau^1 \sim \mu} \left[ \hat{r}(\tau^0) - \hat{r}(\tau^1) - r^t(\tau^0) + r^t(\tau^1) \right] \\
& \leq C_{\text{TR}} \mathbb{E}_{\tau^0, \tau^1 \sim \mu} \left[ |\hat{r}(\tau^0) - \hat{r}(\tau^1) - r^t(\tau^0) + r^t(\tau^1)| \right] \\
& = C_{\text{TR}} \mathcal{E}(r^t; \hat{r}) \leq \lambda \mathcal{E}(r^t; \hat{r})
\end{aligned}
$$

where the last inequality holds since $\mathcal{E}(r^t; \hat{r})$ is non-negative and $\lambda \geq C_{\text{TR}}$. Further, Lemma C.3 implies

$$
\begin{aligned}
\lambda \mathcal{E}(r^t; \hat{r}) & \leq V_{1,r^\star}^{\pi^t} - V_{1,r^\star}^{\mu} - V_{1,r^t}^{\pi^t} + V_{1,r^t}^{\mu} + \lambda \mathcal{E}(r^\star; \hat{r}) + 2\tilde{\epsilon}_{approx} \\
& \leq V_{1,r^\star}^{\pi^t} - V_{1,r^\star}^{\mu} - V_{1,r^t}^{\pi^t} + V_{1,r^t}^{\mu} + \lambda \tilde{\epsilon}_r(\delta/4) + 2\tilde{\epsilon}_{approx}
\end{aligned}
$$

where the last inequality holds due to Lemma E.2:

$$
\begin{aligned}
\mathcal{E}(r^\star; \hat{r}) & = \mathbb{E}_{\tau^0, \tau^1 \sim \mu} \left[ |\hat{r}(\tau^0) - \hat{r}(\tau^1) - r^\star(\tau^0) + r^\star(\tau^1)| \right] \\
& \leq \sqrt{\mathbb{E}_{\tau^0, \tau^1 \sim \mu} \left[ |\hat{r}(\tau^0) - \hat{r}(\tau^1) - r^\star(\tau^0) + r^\star(\tau^1)|^2 \right]} \leq \epsilon_r(\delta/4).
\end{aligned}
$$

Therefore, we have

$$
\text{(II)} \leq \lambda \epsilon_r(\delta/4) + 2\tilde{\epsilon}_{approx}.
$$

**Bounding Sub-optimality.** Putting the bounds on (I) and (II) into (10), we have

$$
\begin{aligned}
& V_{1,r^\star}^{\pi^\star} - V_{1,r^\star}^{\pi^t} \\
& \leq \sqrt{C_{\text{TR}}} \epsilon_r(\delta/4) + \lambda \epsilon_r(\delta/4) + 2\tilde{\epsilon}_{approx} + V_{1,r^t}^{\pi^\star} - V_{1,r^t}^{\pi^t}
\end{aligned}
\tag{11}
$$

Since Algorithm 1 returns the mixture policy $\hat{\pi} = \frac{1}{T} \sum_{t=1}^T \pi^t$, the sub-optimality is $V_{1,r^\star}^{\pi^\star} - V_{1,r^\star}^{\hat{\pi}} = \frac{1}{T} \sum_{t=1}^T \left( V_{1,r^\star}^{\pi^\star} - V_{1,r^\star}^{\pi^t} \right)$. Using the bound in (11) and Lemma C.5, it holds that

$$
\begin{aligned}
& V_{1,r^\star}^{\pi^\star} - V_{1,r^\star}^{\hat{\pi}} \\
& = \frac{1}{T} \sum_{t=1}^T \left( V_{1,r^\star}^{\pi^\star} - V_{1,r^\star}^{\pi^t} \right) \\
& \leq \sqrt{C_{\text{TR}}} \epsilon_r(\delta/4) + \lambda \epsilon_r(\delta/4) + 2\tilde{\epsilon}_{approx} + \frac{1}{T} \sum_{t=1}^T \left( V_{1,r^t}^{\pi^\star} - V_{1,r^t}^{\pi^t} \right) \\
& \leq \sqrt{C_{\text{TR}}} \epsilon_r(\delta/4) + \lambda \epsilon_r(\delta/4) + 2\tilde{\epsilon}_{approx} + RH\sqrt{\frac{\log |\mathcal{A}|}{2T}} + 2H \epsilon_{\text{PE}} \sqrt{C_{\text{ST}}} \\
& \leq \mathcal{O}\left( \sqrt{\log \frac{|\mathcal{R}|}{\delta}} \left( \frac{C_{\text{TR}} \kappa \sqrt{H}}{\sqrt{M}} + \frac{R}{\sqrt{K_1}} + \frac{R}{\sqrt{N}} \right) + RH\sqrt{\frac{\log |\mathcal{A}|}{T}} + RH\sqrt{\frac{C_{\text{ST}}}{K_2} \log \frac{TH|\mathcal{F}|}{\delta}} \right)
\end{aligned}
$$

$\square$

# D    DETAILED PROOF OF THEOREM 4.1

For simplicity, we introduce some notations regarding optimization objectives in Algorithm 2. For $f \in \mathcal{F}^H$, we define

$$\hat{\mathcal{L}}_{\text{opt}}^t(f; \tilde{P}, \tilde{r}) := \sum_{h=1}^{H} \mathbb{E}_{(s_h, a_h) \sim \mathcal{D}_{\text{traj}}} \left[ f_h \circ \pi_h^t(s_h) - f_h(s_h, a_h) \right] + \lambda \hat{\mathcal{E}}_{\mathcal{D}_{\text{traj}}}(f; \tilde{P}, \tilde{r})$$

and its population version as

$$\mathcal{L}_{\text{opt}}^t(f; \tilde{P}, \tilde{r}) := \sum_{h=1}^{H} \mathbb{E}_{(s_h, a_h) \sim d_h^\mu} \left[ f_h \circ \pi_h^t(s_h) - f_h(s_h, a_h) \right] + \lambda \mathcal{E}(f; \tilde{P}, \tilde{r})$$

## D.1    OPTIMIZATION ERROR

In this section, we prove that the (finite-sample) optimization objective $\hat{\mathcal{L}}_{\text{opt}}^t(f; \hat{P}, \hat{r})$ is close to its population version $\mathcal{L}_{\text{opt}}^t(f; \tilde{P}, \tilde{r})$. The result ensures that $f^t$ is a good approximation for the solutions to the optimization program with infinite samples, i.e.

$$f^t \approx \underset{f \in \mathcal{F}^H}{\arg\min} \, \mathcal{L}_{\text{opt}}^t(f; P^\star, \hat{r}).$$

**Remark.** For this section, we assume that the maximum likelihood transition estimator $\hat{P}$ is computed using half of $\mathcal{D}_{\text{traj}}$, and the losses $\hat{\mathcal{L}}_{\text{opt}}^t(f; \hat{P}, \hat{r})$ are computed from the other half. This increases the sample complexity only by a constant factor but helps avoid union bound over $\mathcal{P}$ in the proof of Lemma D.2.

**Lemma D.1.** *With probability at least $1 - \delta/2$, for all $t \in [T]$, we have that*

$$\mathcal{L}_{opt}^t(f^t; \hat{r}) \leq \mathcal{L}_{opt}^t(Q^{\pi^t}; \hat{r}) + 2\epsilon_{approx}$$

*where $\epsilon_{approx}$ is defined in Lemma D.2.*

*Proof.* Consider this decomposition:

$$\mathcal{L}_{\text{opt}}^t(f^t; \hat{r}) - \mathcal{L}_{\text{opt}}^t(Q^{\pi^t}; \hat{r})$$
$$= \underbrace{\mathcal{L}_{\text{opt}}^t(f^t; \hat{r}) - \hat{\mathcal{L}}_{\text{opt}}^t(f^t; \hat{P}, \hat{r})}_{(I)} + \underbrace{\hat{\mathcal{L}}_{\text{opt}}^t(f^t; \hat{P}, \hat{r}) - \hat{\mathcal{L}}_{\text{opt}}^t(Q^{\pi^t}; \hat{P}, \hat{r})}_{(II)} + \underbrace{\hat{\mathcal{L}}_{\text{opt}}^t(Q^{\pi^t}; \hat{P}, \hat{r}) - \mathcal{L}_{\text{opt}}^t(Q^{\pi^t}; \hat{r})}_{(III)}.$$

Conditioned on the event defined by Lemma D.2, (I) and (III) are bounded by $\epsilon_{approx}$. Moreover, the optimality of $f^t$ implies (II)$\leq 0$. $\qquad\square$

**Lemma D.2.** *With probability at least $1 - \delta/2$, for every $t \in [T]$ and $f \in \mathcal{F}^H$, it holds that*

$$\left| \hat{\mathcal{L}}_{opt}^t(f; \hat{P}, \hat{r}) - \mathcal{L}_{opt}^t(f; \hat{r}) \right| \leq 8R\sqrt{\frac{H^3 T \log(8H|\mathcal{F}|/\delta)}{N}} + 2RH\epsilon_P(\delta/8) := \epsilon_{approx}.$$

*Proof.* Due to the policy update in Line 7 of Algorithm 2, the policies $\{\pi_h^t\}_{(t,h) \in [T] \times [H]}$ belongs to the following function class:

$$\Pi = \left\{ \pi(a \mid s) = \frac{\exp\left(\sum_{i=1}^{T} \eta f^i(s, a)\right)}{\sum_{a' \in \mathcal{A}} \exp\left(\sum_{i=1}^{T} \eta f^i(s, a')\right)} : f^i \in \mathcal{F} \text{ for all } i \in [T] \right\}.$$

It is clear that $|\Pi| \leq |\mathcal{F}|^T$.

**Step 1.** Fix $h \in [H]$, $f \in \mathcal{F}$, and $\pi \in \Pi$. Since $|f \circ \pi(s)| \leq R$ for all $s \in \mathcal{S}$, Hoeffding inequality implies that

$$\left| \mathbb{E}_{s_h \in \mathcal{D}_{\text{traj}}} [f \circ \pi(s_h)] - \mathbb{E}_{s_h \sim d_h^\mu} [f \circ \pi(s_h)] \right| \leq R \sqrt{\frac{\log(8/\delta)}{2N}}$$

with probability at least $1 - \delta/8$. Similarly, since $|f(s,a)| \leq R$ for all $(s,a) \in \mathcal{S} \times \mathcal{A}$, it holds that

$$\left| \mathbb{E}_{(s_h,a_h) \sim \mathcal{D}_{\text{traj}}} [f(s_h,a_h)] - \mathbb{E}_{(s_h,a_h) \sim d_h^\mu} [f(s_h,a_h)] \right| \leq R \sqrt{\frac{\log(8/\delta)}{2N}}$$

with probability at least $1 - \delta/8$. Thus, with probability at least $1 - \delta/4$, we have

$$\left| \mathbb{E}_{(s_h,a_h) \sim \mathcal{D}_{\text{traj}}} [f \circ \pi(s_h) - f(s_h,a_h)] - \mathbb{E}_{(s_h,a_h) \sim d_h^\mu} [f \circ \pi(s_h) - f(s_h,a_h)] \right|$$

$$\leq \left| \mathbb{E}_{(s_h,a_h) \sim \mathcal{D}_{\text{traj}}} [f \circ \pi(s_h)] - \mathbb{E}_{(s_h,a_h) \sim d_h^\mu} [f \circ \pi(s_h)] \right|$$

$$+ \left| \mathbb{E}_{(s_h,a_h) \sim \mathcal{D}_{\text{traj}}} [f(s_h,a_h)] - \mathbb{E}_{(s_h,a_h) \sim d_h^\mu} [f(s_h,a_h)] \right|$$

$$\leq R \sqrt{\frac{2\log(8/\delta)}{N}}.$$

Consider union bound over all $h \in [H]$, $f \in \mathcal{F}$, and $\pi \in \Pi$. Since $\pi_h^t \in \Pi$ for every $(t,h) \in [T] \times [H]$, with probability at least $1 - \delta/4$, we have

$$\left| \sum_{h=1}^H \mathbb{E}_{(s_h,a_h) \sim \mathcal{D}_{\text{traj}}} \left[ f_h \circ \pi_h^t(s_h) - f_h(s_h,a_h) \right] - \sum_{h=1}^H \mathbb{E}_{(s_h,a_h) \sim d_h^\mu} \left[ f_h \circ \pi_h^t(s_h) - f_h(s_h,a_h) \right] \right|$$

$$\leq \sum_{h=1}^H \left| \mathbb{E}_{(s_h,a_h) \sim \mathcal{D}_{\text{traj}}} \left[ f_h \circ \pi_h^t(s_h) - f_h(s_h,a_h) \right] - \mathbb{E}_{(s_h,a_h) \sim d_h^\mu} \left[ f_h \circ \pi_h^t(s_h) - f_h(s_h,a_h) \right] \right|$$

$$\leq RH \sqrt{\frac{2\log(8H|\mathcal{F}||\Pi|/\delta)}{N}} \leq 2RH \sqrt{\frac{T\log(8H|\mathcal{F}|/\delta)}{N}}.$$

for every $f \in \mathcal{F}$.

**Step 2.** We have that

$$|\hat{\mathcal{E}}_{\mathcal{D}_{\text{traj}}}(f; \hat{P}, \hat{r}) - \mathcal{E}(f; P^\star, \hat{r})| \leq |\hat{\mathcal{E}}_{\mathcal{D}_{\text{traj}}}(f; \hat{P}, \hat{r}) - \mathcal{E}(f; \hat{P}, \hat{r})| + |\mathcal{E}(f; \hat{P}, \hat{r}) - \mathcal{E}(f; P^\star, \hat{r})|. \quad (12)$$

Again, we use Hoeffding inequality to bound the first term. Fix $f \in \mathcal{F}^H$ and $\pi = \{\pi_h\}_{h=1}^H \in \Pi^H$ and consider the function $r_{\hat{P},f}^\pi$ (Recall that $r_{h,\hat{P},f}^\pi(s,a) = f_h(s,a) - \hat{P}(f_{h+1} \circ \pi_{h+1})(s,a)$ for all $h \in [H]$ and $(s,a) \in \mathcal{S} \times \mathcal{A}$). Since $|(r_{\hat{P},f}^\pi - \hat{r})(\tau)| \leq 2RH$ for any trajectory $\tau$, we have that

$$\left| \mathbb{E}_{(\tau^0,\tau^1) \sim \mathcal{D}_{\text{traj}}} \left[ \left| (r_{\hat{P},f}^\pi - \hat{r})(\tau^0) - (r_{\hat{P},f}^\pi - \hat{r})(\tau^1) \right| \right] - \mathbb{E}_{(\tau^0,\tau^1) \sim \mu} \left[ \left| (r_{\hat{P},f}^\pi - \hat{r})(\tau^0) - (r_{\hat{P},f}^\pi - \hat{r})(\tau^1) \right| \right] \right|$$

$$\leq 2RH \sqrt{\frac{2\log(8/\delta)}{N}}$$

with probability at least $1 - \delta/8$. Applying union bound over all $f \in \mathcal{F}^H$ and $\pi \in \Pi^H$, since $\pi_h^t \in \Pi$ for every $(t,h) \in [T] \times [H]$, it holds that

$$|\hat{\mathcal{E}}_{\mathcal{D}_{\text{traj}}}(f; \hat{P}, \hat{r}) - \mathcal{E}(f; \hat{P}, \hat{r})| \leq 2RH \sqrt{\frac{2H\log(8|\mathcal{F}||\Pi|/\delta)}{N}} \leq 4RH \sqrt{\frac{HT\log(8|\mathcal{F}|/\delta)}{N}} \quad (13)$$

for every $f \in \mathcal{F}$, with probability at least $1 - \delta/8$.

On the other hand, the second term in (12) is bounded by

$$|\mathcal{E}(f; \hat{P}, \hat{r}) - \mathcal{E}(f; P^\star, \hat{r})|$$

$$\leq \mathbb{E}_{(\tau^0, \tau^1) \sim \mu} \left[ \left| \sum_{h=1}^{H} (P^\star - \hat{P})(f_h \circ \pi_h^t)(s_h^0, a_h^0) - \sum_{h=1}^{H} (P^\star - \hat{P})(f_h \circ \pi_h^t)(s_h^1, a_h^1) \right| \right]$$

$$\leq \mathbb{E}_{\tau^0 \sim \mu} \left[ \sum_{h=1}^{H} \left| (P^\star - \hat{P})(f_h \circ \pi_h^t)(s_h^0, a_h^0) \right| \right] + \mathbb{E}_{\tau^1 \sim \mu} \left[ \sum_{h=1}^{H} \left| (P^\star - \hat{P})(f_h \circ \pi_h^t)(s_h^1, a_h^1) \right| \right]$$

$$= 2 \mathbb{E}_{\tau \sim \mu} \left[ \sum_{h=1}^{H} \left| (P^\star - \hat{P})(f_h \circ \pi_h^t)(s_h, a_h) \right| \right]$$

$$\leq 2R \mathbb{E}_{\tau \sim \mu} \left[ \sum_{h=1}^{H} \left\| P^\star(\cdot \mid s_h, a_h) - \hat{P}(\cdot \mid s_h, a_h) \right\|_1 \right]$$

$$= 2R \sum_{h=1}^{H} \mathbb{E}_{(s_h, a_h) \sim d_h^\mu} \left[ \left\| P^\star(\cdot \mid s_h, a_h) - \hat{P}(\cdot \mid s_h, a_h) \right\|_1 \right]$$

where the first inequality holds since we have $||a| - |b|| \leq |a - b|$ for all $a, b \in \mathbb{R}$, and the third inequality holds due to Hölder's inequality with the fact that $\|f_h \circ \pi_h^t\|_\infty \leq R$. Furthermore, Lemma E.3 implies

$$|\mathcal{E}(f; \hat{P}, \hat{r}) - \mathcal{E}(f; P^\star, \hat{r})| \leq 2R \sum_{h=1}^{H} \mathbb{E}_{(s_h, a_h) \sim d_h^\mu} \left[ \left\| P^\star(\cdot \mid s_h, a_h) - \hat{P}(\cdot \mid s_h, a_h) \right\|_1 \right]$$

$$\leq 2R \sum_{h=1}^{H} \sqrt{ \mathbb{E}_{(s_h, a_h) \sim d_h^\mu} \left[ \left\| P^\star(\cdot \mid s_h, a_h) - \hat{P}(\cdot \mid s_h, a_h) \right\|_1^2 \right] }$$

$$\leq 2RH \epsilon_P(\delta/8) \tag{14}$$

with probability at least $1 - \delta/8$. Taking union bound of the two event (13) and (14), with probability at least $1 - \delta/4$, it holds that

$$|\hat{\mathcal{E}}_{\mathcal{D}_{\text{traj}}}(f; \hat{P}, \hat{r}) - \mathcal{E}(f; P^\star, \hat{r})| \leq 2RH \sqrt{\frac{HT \log(8|\mathcal{F}|/\delta)}{N}} + 2RH \epsilon_P(\delta/8)$$

for every $f \in \mathcal{F}$.

Finally, we conclude the proof by combining the bounds in Step 1 and Step 2. With probability at least $1 - \delta/2$, for every $f \in \mathcal{F}$, it hols that

$$\left| \hat{\mathcal{L}}_{\text{opt}}^t(f; \hat{P}, \hat{r}) - \mathcal{L}_{\text{opt}}^t(f; \hat{r}) \right|$$

$$\leq \left| \sum_{h=1}^{H} \mathbb{E}_{(s_h, a_h) \sim \mathcal{D}_{\text{traj}}} \left[ f_h \circ \pi_h^t(s_h) - f_h(s_h, a_h) \right] - \sum_{h=1}^{H} \mathbb{E}_{(s_h, a_h) \sim d_h^\mu} \left[ f_h \circ \pi_h^t(s_h) - f_h(s_h, a_h) \right] \right|$$

$$+ \left| \hat{\mathcal{E}}_{\mathcal{D}_{\text{traj}}}(f; \hat{P}, \hat{r}) - \mathcal{E}(f; P^\star, \hat{r}) \right|$$

$$\leq 4RH \sqrt{\frac{T \log(8H|\mathcal{F}|/\delta)}{N}} + 2RH \sqrt{\frac{HT \log(8|\mathcal{F}|/\delta)}{N}} + 2RH \epsilon_P(\delta/8)$$

$$\leq 8R \sqrt{\frac{H^3 T \log(8H|\mathcal{F}|/\delta)}{N}} + 2RH \epsilon_P(\delta/8).$$

$\square$

### D.2 POLICY UPDATE

The analysis of the policy update step in Algorithm 2 follows the same argument in Lemma C.5.

**Lemma D.3.** *For any sequence of functions $\{f^t\}_{t=1}^T$, the policy update (Line 7) in Algorithm 2 guarantees that*

$$\frac{1}{T}\sum_{t=1}^T\left(V_{1,r^t}^{\pi^\star}(s_1) - V_{1,r^t}^{\pi^t}(s_1)\right) \le RH\sqrt{\frac{\log|\mathcal{A}|}{2T}}$$

*where $r^t = r_{P^\star,f^t}^{\pi^t}$, i.e. $r_h^t(s,a) = f_h^t(s,a) - P^\star(f_{h+1}^t \circ \pi_{h+1}^t)(s,a)$ for all $h \in [H]$ and $(s,a) \in \mathcal{S} \times \mathcal{A}$.*

*Proof.* Since we have the Bellman equation $f_h^t = r_h^t + P_h^\star(f_{h+1}^t \circ \pi_{h+1}^t)$ for all $h \in [H]$, we can apply the performance difference lemma (Lemma E.1) to obtain

$$\sum_{t=1}^T\left(V_{1,r^t}^{\pi^\star}(s_1) - V_{1,r^t}^{\pi^t}(s_1)\right) = \sum_{t=1}^T\sum_{h=1}^H \mathbb{E}_{\pi^\star}\left[\langle f_h^t(s_h,\cdot), \pi_h^\star(\cdot \mid s_h) - \pi_h^t(\cdot \mid s_h)\rangle\right].$$

Rearranging the inner product term, we see that

$$\langle \eta f_h^t(s_h,\cdot), \pi_h^\star(\cdot \mid s) - \pi_h^t(\cdot \mid s)\rangle$$
$$\langle \eta f_h^t(s_h,\cdot), \pi_h^\star(\cdot \mid s) - \pi_h^{t+1}(\cdot \mid s)\rangle + \langle \eta f_h^t(s_h,\cdot), \pi_h^{t+1}(\cdot \mid s) - \pi_h^t(\cdot \mid s)\rangle$$
$$\le \langle \eta f_h^t(s_h,\cdot), \pi_h^\star(\cdot \mid s) - \pi_h^{t+1}(\cdot \mid s)\rangle + \eta \left\|f_h^t(s_h,\cdot)\right\|_\infty \left\|\pi_h^\star(\cdot \mid s) - \pi_h^{t+1}(\cdot \mid s)\right\|_1$$
$$\le \langle \eta f_h^t(s_h,\cdot), \pi_h^\star(\cdot \mid s) - \pi_h^{t+1}(\cdot \mid s)\rangle + \eta R \left\|\pi_h^\star(\cdot \mid s) - \pi_h^{t+1}(\cdot \mid s)\right\|_1 \tag{15}$$

where we use Hölder's inequality with the fact that $\|f_h^t\|_\infty \le R$. Now recall that the policy update step in Algorithm 3 leads to

$$\pi_h^{t+1}(\cdot \mid s) = \frac{1}{Z_h^t(s)}\pi_h^t(\cdot \mid s)\exp\left(\eta f_h^t(s,\cdot)\right)$$

where $Z_h^t(s) = \sum_{a\in\mathcal{A}}\pi_h^t(a \mid s)\exp\left(\eta f_h^t(s,a)\right)$. Using the relationship $\eta f_h^t(s,a) = \log Z_h^t(s) + \log\pi_h^{t+1}(a \mid s) - \log\pi_h^t(a \mid s)$, it holds that

$$\langle \eta f_h^t(s_h,\cdot), \pi_h^\star(\cdot \mid s) - \pi_h^{t+1}(\cdot \mid s)\rangle$$
$$= \langle \log Z_h^t(s) + \log\pi_h^{t+1}(\cdot \mid s) - \log\pi_h^t(\cdot \mid s), \pi_h^\star(\cdot \mid s) - \pi_h^{t+1}(\cdot \mid s)\rangle$$
$$= \langle \log\pi_h^{t+1}(\cdot \mid s) - \log\pi_h^t(\cdot \mid s), \pi^\star(\cdot \mid s) - \pi_h^{t+1}(\cdot \mid s)\rangle$$
$$= \langle \log\pi_h^{t+1}(\cdot \mid s) - \log\pi_h^t(\cdot \mid s), \pi^\star(\cdot \mid s)\rangle - D_{KL}\left(\pi_h^{t+1}(\cdot \mid s)\|\pi_h^t(\cdot \mid s)\right)$$
$$= \langle \log\frac{\pi_h^{t+1}(\cdot \mid s)}{\pi_h^\star(\cdot \mid s)} + \log\frac{\pi_h^\star(\cdot \mid s)}{\pi_h^t(\cdot \mid s)}, \pi_h^\star(\cdot \mid s)\rangle - D_{KL}\left(\pi_h^{t+1}(\cdot \mid s)\|\pi_h^t(\cdot \mid s)\right)$$
$$= D_{KL}\left(\pi_h^\star(\cdot \mid s)\|\pi_h^t(\cdot \mid s)\right) - D_{KL}\left(\pi_h^\star(\cdot \mid s)\|\pi_h^{t+1}(\cdot \mid s)\right) - D_{KL}\left(\pi_h^{t+1}(\cdot \mid s)\|\pi_h^t(\cdot \mid s)\right)$$
$$\le D_{KL}\left(\pi_h^\star(\cdot \mid s)\|\pi_h^t(\cdot \mid s)\right) - D_{KL}\left(\pi_h^\star(\cdot \mid s)\|\pi_h^{t+1}(\cdot \mid s)\right) - \frac{1}{2}\left\|\pi_h^\star(\cdot \mid s) - \pi_h^{t+1}(\cdot \mid s)\right\|_1^2$$

where the second equality holds since $Z_h^t(s)$ is a constant given $s$, and the last inequality holds due to Pinsker's inequality. Combining this bound with (15), we obtain

$$\sum_{t=1}^{T} \langle \eta f_h^t(s_h, \cdot), \pi_h^\star(\cdot \mid s) - \pi_h^t(\cdot \mid s) \rangle$$

$$= \sum_{t=1}^{T} \left( D_{KL} \left( \pi_h^\star(\cdot \mid s) \| \pi_h^t(\cdot \mid s) \right) - D_{KL} \left( \pi_h^\star(\cdot \mid s) \| \pi_h^{t+1}(\cdot \mid s) \right) \right)$$

$$+ \sum_{t=1}^{T} \left( \eta R \left\| \pi_h^\star(\cdot \mid s) - \pi_h^{t+1}(\cdot \mid s) \right\|_1 - \frac{1}{2} \left\| \pi_h^\star(\cdot \mid s) - \pi_h^{t+1}(\cdot \mid s) \right\|_1^2 \right)$$

$$\leq \sum_{t=1}^{T} \left( D_{KL} \left( \pi_h^\star(\cdot \mid s) \| \pi_h^t(\cdot \mid s) \right) - D_{KL} \left( \pi_h^\star(\cdot \mid s) \| \pi_h^{t+1}(\cdot \mid s) \right) \right) + \sum_{t=1}^{T} \frac{\eta^2 R^2}{2}$$

$$= D_{KL} \left( \pi_h^\star(\cdot \mid s) \| \pi_h^1(\cdot \mid s) \right) - D_{KL} \left( \pi_h^\star(\cdot \mid s) \| \pi_h^{T+1}(\cdot \mid s) \right) + \frac{\eta^2 R^2 T}{2}$$

$$\leq \log |\mathcal{A}| + \frac{\eta^2 R^2 T}{2}$$

where the first inequality holds since $\forall x \in \mathbb{R} \ ax - x^2/2 \leq a^2/2$, and the second inequality holds due to the fact that $\pi_h^1 = \mathrm{Unif}(\mathcal{A})$. Finally, setting $\eta = \sqrt{\frac{2 \log |\mathcal{A}|}{R^2 T}}$, we have

$$\sum_{t=1}^{T} V_{1,r^t}^{\pi^\star} - V_{1,r^t}^{\pi^t} = \sum_{t=1}^{T} \mathbb{E}_{\pi^\star} \left[ \sum_{h=1}^{H} \langle f_h^t(s_h, \cdot), \pi_h^\star(\cdot \mid s_h) - \pi_h^t(\cdot \mid s_h) \rangle \right]$$

$$= \sum_{h=1}^{H} \mathbb{E}_{\pi^\star} \left[ \sum_{t=1}^{T} \langle f_h^t(s_h, \cdot), \pi_h^\star(\cdot \mid s_h) - \pi_h^t(\cdot \mid s_h) \rangle \right]$$

$$\leq \sum_{h=1}^{H} \left( \frac{\log |\mathcal{A}|}{\eta} + \frac{\eta R^2 T}{2} \right) = RH \sqrt{\frac{T \log |\mathcal{A}|}{2}}.$$

$\square$

Finally, we prove Theorem 4.1.

*Proof of Theorem 4.1.* For simplicity, we write $r^t = r_{P^\star, f^t}^{\pi^t}$, i.e. $r_h^t(s, a) = f_h^t(s, a) - P_h^\star(f_{h+1}^t \circ \pi_{h+1}^t)(s, a)$ for all $(s, a) \in \mathcal{S} \times \mathcal{A}$ and $h \in [H]$. The condition $r^t \in \mathcal{R}^H$ is not required; we only rely on the boundedness $\|r_h^t\|_\infty \leq R$ for all $h$, which Assumption 3 guarantees.

Condition on the events in Lemma E.2 (with $\delta' = \delta/2$) and Lemma D.1, which hold simultaneously with probability at least $1 - \delta$. Consider the following sub-optimality decomposition at step $t$:

$$V_{1,r^\star}^{\pi^\star} - V_{1,r^\star}^{\pi^t} = V_{1,r^\star}^{\pi^\star} - V_{1,\hat{r}}^{\pi^\star} + V_{1,\hat{r}}^{\pi^\star} - V_{1,r^t}^{\pi^\star} + V_{1,r^t}^{\pi^\star} - V_{1,r^\star}^{\pi^t} + V_{1,r^t}^{\pi^t} - V_{1,r^t}^{\pi^t}$$

$$= \underbrace{V_{1,r^\star - \hat{r}}^{\pi^\star} - V_{1,r^\star - \hat{r}}^{\mu}}_{\text{(I) : MLE estimation error}}$$

$$+ \underbrace{V_{1,\hat{r} - r^t}^{\pi^\star} - V_{1,\hat{r} - r^t}^{\mu} - V_{1,r^\star}^{\pi^t} + V_{1,r^\star}^{\mu} + V_{1,r^t}^{\pi^t} - V_{1,r^t}^{\mu}}_{\text{(II) : Optimization error}}$$

$$+ \underbrace{V_{1,r^t}^{\pi^\star} - V_{1,r^t}^{\pi^t}}_{\text{(III) : Policy update regret}}, \tag{16}$$

where we omit the initial state $s_1$ for simplicity.

**Bounding (I).** Using Lemma E.2, the MLE estimation error is bounded by:

$$
\begin{aligned}
\text{(I)} &= V_{1,r^\star-\hat{r}}^{\pi^\star} - V_{1,r^\star-\hat{r}}^{\mu} \\
&= \mathbb{E}_{\tau^0\sim\pi^\star,\tau^1\sim\mu}\left[r^\star(\tau^0)-r^\star(\tau^1)-\hat{r}(\tau^0)+\hat{r}(\tau^1)\right] \\
&\leq \sqrt{\mathbb{E}_{\tau^0\sim\pi^\star,\tau^1\sim\mu}\left[|r^\star(\tau^0)-r^\star(\tau^1)-\hat{r}(\tau^0)+\hat{r}(\tau^1)|^2\right]} \\
&\leq \sqrt{C_{\mathrm{TR}}\mathbb{E}_{\tau^0,\tau^1\sim\mu}\left[|r^\star(\tau^0)-r^\star(\tau^1)-\hat{r}(\tau^0)+\hat{r}(\tau^1)|^2\right]} \\
&\leq \sqrt{C_{\mathrm{TR}}}\epsilon_r(\delta/2)
\end{aligned}
$$

**Bounding (II).** We can relate the terms $V_{1,\hat{r}-r^t}^{\pi^\star}-V_{1,\hat{r}-r^t}^{\mu}$ to the trajectory-pair $\ell_1$ loss $\mathcal{E}(f^t;P^\star,\hat{r})$. By Assumption 4, we have that

$$
\begin{aligned}
&V_{1,\hat{r}-r^t}^{\pi^\star} - V_{1,\hat{r}-r^t}^{\mu} \\
&= \mathbb{E}_{\tau^0\sim\pi^\star,\tau^1\sim\mu}\left[\hat{r}(\tau^0)-\hat{r}(\tau^1)-r^t(\tau^0)+r^t(\tau^1)\right] \\
&\leq C_{\mathrm{TR}}\mathbb{E}_{\tau^0,\tau^1\sim\mu}\left[|\hat{r}(\tau^0)-\hat{r}(\tau^1)-r^t(\tau^0)+r^t(\tau^1)|\right] \\
&= C_{\mathrm{TR}}\mathbb{E}_{\tau^0,\tau^1\sim\mu}\left[|r_{P^\star,f^t}^{\pi^t}(\tau^0)-r_{P^\star,f^t}^{\pi^t}(\tau^1)-\hat{r}(\tau^0)+\hat{r}(\tau^1)|\right] \\
&= C_{\mathrm{TR}}\mathcal{E}(f^t;P^\star,\hat{r}) \leq \lambda\mathcal{E}(f^t;P^\star,\hat{r})
\end{aligned}
$$

where the last inequality holds since $\mathcal{E}(f^t;P^\star,\hat{r})$ is non-negative and $\lambda \geq C_{\mathrm{TR}}$. Further, Lemma D.1 and Lemma E.1 implies

$$
\begin{aligned}
\lambda\mathcal{E}(f^t;P^\star,\hat{r}) &\leq \sum_{h=1}^{H}\mathbb{E}_{(s_h,a_h)\sim d_h^\mu}\left[Q_h^{\pi^t}\circ\pi_h^t(s_h)-Q_h^{\pi^t}(s_h,a_h)\right] + \lambda\mathcal{E}(Q^{\pi^t};P^\star,\hat{r}) \\
&\quad - \sum_{h=1}^{H}\mathbb{E}_{(s_h,a_h)\sim d_h^\mu}\left[f_h^t\circ\pi_h^t(s_h)-f_h^t(s_h,a_h)\right] + 2\epsilon_{approx} \\
&= \left(V_{1,r^\star}^{\pi^t}-V_{1,r^\star}^{\mu}\right) + \lambda\mathcal{E}(Q^{\pi^t};P^\star,\hat{r}) - \left(V_{1,r^t}^{\pi^t}-V_{1,r^t}^{\mu}\right) + 2\epsilon_{approx}.
\end{aligned}
$$

On the other hand, note that

$$
\begin{aligned}
r_{P^\star,Q^{\pi^t}}^{\pi^t}(\tau) &= \sum_{h=1}^{H}\left(Q_h^{\pi^t}(s_h,a_h)-P^\star(Q_{h+1}^{\pi^t}\circ\pi_{h+1}^t)(s_h,a_h)\right) \\
&= \sum_{h=1}^{H}\left(Q_h^{\pi^t}(s_h,a_h)-P^\star V_{h+1}^{\pi^t}(s_h,a_h)\right) \\
&= \sum_{h=1}^{H}r_h^\star(s_h,a_h) = r^\star(\tau)
\end{aligned}
$$

for any $\tau=(s_1,a_1,\ldots,s_H,a_H)$, i.e. $r_{P^\star,Q^{\pi^t}}^{\pi^t}=r^\star$. Thus, we have

$$
\begin{aligned}
\lambda\mathcal{E}(Q^{\pi^t};P^\star,\hat{r}) &= \lambda\mathbb{E}_{(\tau_0,\tau_1)\sim\mu}\left[|\{r^\star(\tau^0)-r^\star(\tau^1)\}-\{\hat{r}(\tau^0)-\hat{r}(\tau^1)\}|\right] \\
&\leq \lambda\sqrt{\mathbb{E}_{(\tau_0,\tau_1)\sim\mu}\left[|\{r^\star(\tau^0)-r^\star(\tau^1)\}-\{\hat{r}(\tau^0)-\hat{r}(\tau^1)\}|^2\right]} \leq \lambda\epsilon_r(\delta/2)
\end{aligned}
$$

where the inequality follows from Lemma E.2. Combining the results, we obtain

$$
\begin{aligned}
\text{(II)} &= \left(V_{1,\hat{r}-r^t}^{\pi^\star}-V_{1,\hat{r}-r^t}^{\mu}\right) - \left(V_{1,r^\star}^{\pi^t}-V_{1,r^\star}^{\mu}\right) + \left(V_{1,r^t}^{\pi^t}-V_{1,r^t}^{\mu}\right) \\
&\leq \lambda\mathcal{E}(Q^{\pi^t};P^\star,\hat{r}) + 2\epsilon_{approx} \leq \lambda\epsilon_r(\delta/2) + 2\epsilon_{approx}
\end{aligned}
$$

**Bounding Sub-optimality.** Finally, we bound the sub-optimality $V_{1,r^\star}^{\pi^\star} - V_{1,r^\star}^{\hat{\pi}}$. Putting the bounds on (I) and (II) into (16), we have

$$V_{1,r^\star}^{\pi^\star} - V_{1,r^\star}^{\pi^t}$$
$$\leq \sqrt{C_{\text{TR}}}\epsilon_r(\delta/2) + \lambda\epsilon_r(\delta/2) + 2\epsilon_{approx} + V_{1,r^t}^{\pi^\star} - V_{1,r^t}^{\pi^t}$$

Since Algorithm 2 returns the mixture policy $\hat{\pi} = \frac{1}{T}\sum_{t=1}^{T}\pi^t$, the sub-optimality is $V_{1,r^\star}^{\pi^\star} - V_{1,r^\star}^{\hat{\pi}} = \frac{1}{T}\sum_{t=1}^{T}\left(V_{1,r^\star}^{\pi^\star} - V_{1,r^\star}^{\pi^t}\right)$. Using the bounds we derived and Lemma D.3, it holds that

$$V_{1,r^\star}^{\pi^\star} - V_{1,r^\star}^{\hat{\pi}}$$

$$= \frac{1}{T}\sum_{t=1}^{T}\left(V_{1,r^\star}^{\pi^\star} - V_{1,r^\star}^{\pi^t}\right)$$

$$\leq \sqrt{C_{\text{TR}}}\epsilon_r(\delta/2) + \lambda\epsilon_r(\delta/2) + 2\epsilon_{approx} + \frac{1}{T}\sum_{t=1}^{T}\left(V_{1,r^t}^{\pi^\star} - V_{1,r^t}^{\pi^t}\right)$$

$$\leq \sqrt{C_{\text{TR}}}\epsilon_r(\delta/2) + \lambda\epsilon_r(\delta/2) + 2\epsilon_{approx} + RH\sqrt{\frac{\log|\mathcal{A}|}{2T}}$$

$$\leq \mathcal{O}\left(C_{\text{TR}}\sqrt{\frac{\kappa^2 H\log(|\mathcal{R}|/\delta)}{M}} + R\sqrt{\frac{H^3 T\log(H|\mathcal{F}|/\delta)}{N}} + RH\sqrt{\frac{\log(H|\mathcal{P}|/\delta)}{N}} + RH\sqrt{\frac{\log|\mathcal{A}|}{T}}\right)$$

$$\leq \mathcal{O}\left(C_{\text{TR}}\sqrt{\frac{\kappa^2 H\log(|\mathcal{R}|/\delta)}{M}} + RH\sqrt{\frac{\max\{HT\log(H|\mathcal{F}|/\delta), \log(H|\mathcal{P}|/\delta)\}}{N}} + RH\sqrt{\frac{\log|\mathcal{A}|}{T}}\right).$$

$\square$

# E    SUPPORTING LEMMAS

**Lemma E.1** (Performance Difference Lemma (Kakade & Langford, 2002)). *Let $P$ be any transition probability, and denote the corresponding value function by $V$. Let $\pi, \tilde{pi}$ be any policies. For any reward $r$, we have that*

$$V_{1,r}^{\pi}(s_1) - V_{1,r}^{\tilde{\pi}}(s_1) = \sum_{h=1}^{H}\mathbb{E}_{s_h\sim d_h^{\pi}}\left[\langle Q_{h,r}^{\tilde{\pi}}(s_h,\cdot), \pi(\cdot\mid s_h) - \tilde{\pi}(\cdot\mid s_h)\rangle\right]$$

*Proof.* Recursively applying the Bellman equation, we obtain

$$V_{1,r}^{\pi}(s_1) - V_{1,r}^{\tilde{\pi}}(s_1) = \mathbb{E}_{\pi}[r(s_1,a_1) + V_{2,r}^{\pi}(s_2)] - \mathbb{E}_{\pi}[V_{1,r}^{\tilde{\pi}}(s_1)]$$
$$= \mathbb{E}_{\pi}[Q_{1,r}^{\tilde{\pi}}(s_1,a_1) - V_{2,r}^{\tilde{\pi}}(s_2) + V_{2,r}^{\pi}(s_2)] - \mathbb{E}_{\pi}[V_{1,r}^{\tilde{\pi}}(s_1)]$$
$$= \mathbb{E}_{\pi}[Q_{1,r}^{\tilde{\pi}}(s_1,a_1) - V_{1,r}^{\tilde{\pi}}(s_1)] + \mathbb{E}_{\pi}[V_{2,r}^{\pi}(s_2) - V_{2,r}^{\tilde{\pi}}(s_2)]$$
$$= \mathbb{E}_{\pi}[\langle Q_{1,r}^{\tilde{\pi}}(s_1,\cdot), \pi(\cdot\mid s_1) - \tilde{\pi}(\cdot\mid s_1)\rangle] + \mathbb{E}_{\pi}[V_{2,r}^{\pi}(s_2) - V_{2,r}^{\tilde{\pi}}(s_2)]$$
$$= \cdots$$
$$= \sum_{h=1}^{H}\mathbb{E}_{s_h\sim d_h^{\pi}}\left[\langle Q_{h,r}^{\tilde{\pi}}(s_h,\cdot), \pi(\cdot\mid s_h) - \tilde{\pi}(\cdot\mid s_h)\rangle\right].$$

$\square$

**Lemma E.2** (Lemma 2 in Zhan et al. (2024a)). *With probability at least $1 - \delta'$, we have*

$$\mathbb{E}_{\tau^0,\tau^1\sim\mu}\left[|(\hat{r}(\tau^0) - \hat{r}(\tau^1)) - (r^\star(\tau^0) - r^\star(\tau^1))|^2\right] \leq \frac{c_1\kappa^2 H\log(|\mathcal{R}|/\delta')}{M} := \epsilon_r^2(\delta')$$

**Lemma E.3** (Lemma 3 in Zhan et al. (2024a)). *With probability at least $1 - \delta'$, for all $h \in [H]$, it holds that*

$$\mathbb{E}_{(s_h,a_h)\sim d_h^\mu}\left[\left\|\hat{P}_h(\cdot \mid s_h, a_h) - P^\star(\cdot \mid s_h, a_h)\right\|_1^2\right] \leq \frac{c_2 \log(H|\mathcal{P}|/\delta')}{N} := \epsilon_P^2(\delta')$$

*where $c_2$ is an absolute constant.*

**Lemma E.4** (Lemma 15 in Song et al. (2023)). *Fix any $B > 0$, $\delta \in (0, 1)$ and assume we have a class of real-valued functions $\mathcal{H} : \mathcal{X} \to [-B, B]$. Suppose we have $K$ i.i.d. samples $\{(x_k, y_k)\}_{k=1}^K$ where $x_k \sim \rho$ and $y_k = h^\star(x_k) + \epsilon_k$ where $h^\star \in \mathcal{H}$ and $\{\epsilon_k\}_{k=1}^K$ are independent random variables such that $\mathbb{E}[\epsilon_k \mid x_k] = 0$. Additionally, suppose that $\max_k |y_k| \leq R$ and $\sup_{x \in \mathcal{X}} |h^\star(x)| \leq B$. Then, with probability at least $1 - \delta$, the least square estimator $\hat{h} \in \arg\min_{h \in \mathcal{H}} \sum_{k=1}^K (h(x_k) - y_k)^2$ satisfies:*

$$\mathbb{E}_{x \sim \rho}\left[\left(\hat{h}(x) - h^\star(x)\right)^2\right] \leq \frac{c_2 B^2 \log(|\mathcal{H}|/\delta)}{K}$$

*where $c_2$ is an absolute constant.*

---

**Algorithm 4** APPO (Practical version)

---

1: **Input:** Batch size $B$, Learning rates $\alpha_\phi, \alpha_\psi, \alpha_\theta$, constants $\lambda > 0$, $\tau \in (0, 1)$
2: Train reward model $\hat{r}$ based on $\mathcal{D}_{\text{pref}}$            ▷ Use any reward learning method
3: **for** step$= 1, 2, \dots$ **do**
4:     Sample mini-batch of transition tuples $\mathcal{B}_{\text{tup}}$ and trajectory pairs $\mathcal{B}_{\text{traj}}$ from $\mathcal{D}_{\text{traj}}$
5:     Train Q functions $\phi^i \leftarrow \phi^i - \alpha_\phi \nabla_{\phi^i} \mathcal{L}_{\phi^i}^\lambda(\mathcal{B}_{\text{tup}}, \mathcal{B}_{\text{traj}})$ for $i \in \{1, 2\}$ (6)
6:     Update target Q function $\bar{\phi}^i = (1 - \tau)\bar{\phi}^i + \tau\phi^i$ for $i \in \{1, 2\}$
7:     Train V function $\psi \leftarrow \psi - \alpha_\psi \nabla_\psi \mathcal{L}_\psi(\mathcal{B}_{\text{tup}})$ (7)
8:     Train actor $\theta \leftarrow \theta + \alpha_\theta \nabla_\theta \mathcal{L}_\theta(\mathcal{B}_{\text{tup}})$ (8)
9: **end for**

---

## F    ADDITIONAL EXPERIMENTS

### F.1    EVALUATION ON THE META-WORLD `medium-expert` DATASET

To further assess the generalization capability of `APPO`, we collected the Meta-World `medium-expert` dataset following the data collection procedures outlined in prior works (Hejna & Sadigh, 2024; Choi et al., 2024). Detailed information on the dataset is provided in Section G. For comparison, we use MR, the most effective baseline method identified in Table 1. The results in Table 2 show that `APPO` consistently outperforms or matches MR.

| # of feedback | 500 | | 1000 | |
|---|---|---|---|---|
| Dataset | dial-turn | sweep-into | dial-turn | sweep-into |
| MR | $15.80_{\pm 12.73}$ | $14.32_{\pm 3.39}$ | $26.08_{\pm 18.78}$ | $8.48_{\pm 1.92}$ |
| APPO | $32.40_{\pm 13.56}$ | $12.80_{\pm 5.35}$ | $39.20_{\pm 15.69}$ | $14.56_{\pm 6.25}$ |

Table 2: Success rates on Meta-World `medium-expert` dataset with 500, 1000 preference feedback samples, averaged over 5 random seeds.

### F.2    EFFECT OF DATASET SIZE

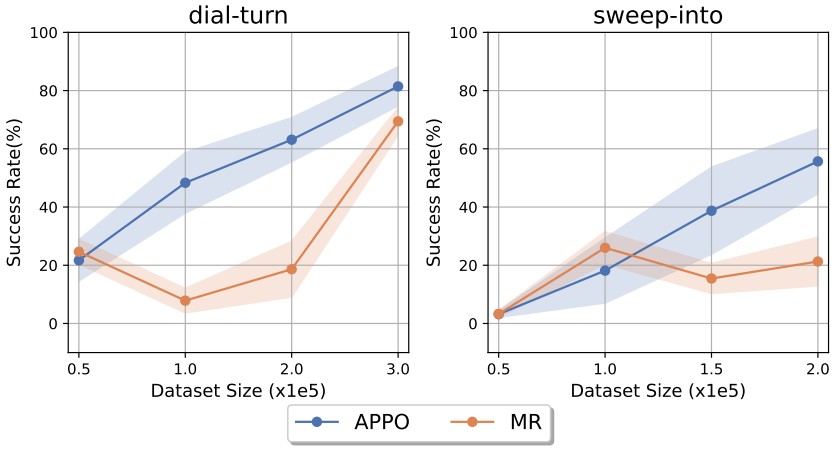

Figure 3: Success rates of `APPO` and MR evaluated in Meta-World `medium-replay` datasets, with varying dataset sizes. The number of preference feedback samples is fixed at 1000.

To examine the impact of dataset size $|\mathcal{D}_{\text{traj}}|$, we conducted experiments with varying sizes of the Meta-World `medium-replay` datasets. As shown in Figure 3, the performance of MR fluctuates with changes in dataset size, whereas the performance of `APPO` exhibits a more consistent and gradual response to dataset size variations.

### F.3    LEARNING CURVES FROM EXPERIMENTS.

Figure 4 and Figure 5 show the learning curves of the experiments in Table 1 and Table 2. Each algorithm is trained for 250,000 gradient steps, with evaluations conducted every 5,000 steps. The success rates from the final five evaluation points are averaged and reported in Table 1 and Table 2.

| Dataset | BPT | box-close | dial-turn | sweep | BPT-wall | sweep-into | drawer-open | lever-pull |
|---|---|---|---|---|---|---|---|---|
| Size ($\times 10^5$) | 1.0 | 8.0 | 3.0 | 7.0 | 1.5 | 1.0 | 1.0 | 3.0 |

Table 3: The sizes of Meta-World `medium-replay` datasets (Choi et al., 2024). The abbreviation BPT indicates button-press-topdown.

## G EXPERIMENTAL DETAILS

### G.1 DATASETS

The Meta-World `medium-replay` dataset from Choi et al. (2024) consists of replay buffers generated by SAC (Haarnoja et al., 2018) agents with an approximate success rate of 50%. The dataset sizes are detailed in Table 3.

The Meta-World `medium-expert` dataset was collected following the procedures described in prior works (Hejna & Sadigh, 2024; Choi et al., 2024). Each dataset contains trajectories from five sources: (1) an expert policy, (2) expert policies for randomized variants and goals of the task, (3) expert policies for different tasks, (4) a random policy, and (5) an $\epsilon$-greedy expert policy that takes greedy actions with a 50% probability. These trajectories are included in the dataset in proportions of $1 : 1 : 2 : 4 : 4$, respectively. Additionally, standard Gaussian noise was added to the actions of each policy. The dataset sizes match those of the `medium-replay` dataset.

### G.2 IMPLEMENTATION AND HYPERPARAMETERS.

For a fair comparison with baseline methods, we train the reward model and MR following the official implementation of Choi et al. (2024). The reward model is implemented by an ensemble model of three fully connected neural networks with three hidden layers, each containing 128 neurons. For critics (Q and V) and policies, we use fully connected neural networks with three hidden layers of 256 neurons each. Other hyperparameters are listed in Table 4. We find that using a lower learning rate for $\pi$ and softer target network updates improves the stability of `APPO` training. Experiments were conducted on an Intel Xeon Gold 6226R CPU and an Nvidia GeForce RTX 3090 GPU. Each training session consists of 250,000 gradient steps, taking approximately 3-4 hours to complete. Our code is available at `https://github.com/oh-lab/APPO.git`.

| Algorithm | Component | Value |
|---|---|---|
| Reward model | Neural networks | 3-layers, hidden dimension 128 |
| | Activation | ReLU for hidden activations, Tanh for final activation |
| | Optimizer | Adam (Kingma & Ba, 2015) with learning rate 1e-3 |
| | Batch size | 512 |
| | Epochs | 300 |
| | Ensembles | 3 |
| MR | Neural networks (Q, V, $\pi$) | 3-layers, hidden dimension 256 |
| | Activaton | ReLU for hidden activations |
| | Q, V, $\pi$ optimizer | Adam with learning rate 3e-4 |
| | Batch size | 256 |
| | Target network soft update | 0.005 |
| | $\beta$ (IQL advantage weight) | 3.0 |
| | $\tau$ (IQL expectile parameter) | 0.7 |
| | discount factor | 0.99 |
| APPO | Neural networks (Q, V, $\pi$) | 3-layers, hidden dimension 256 |
| | Activaton | LeakyReLU for hidden activations |
| | Q,V, $\alpha$ optimizer | Adam with learning rate 3e-4 |
| | $\pi$ optimizer | Adam with learning rate 3e-5 |
| | Batch size | 256 transitions and 16 trajectory pairs |
| | Target network soft update | 0.001 |
| | discount factor | 0.99 |

Table 4: Implementation details and hyperparameters. For the reward model and MR algorithm, we follow the official implementation of Choi et al. (2024).

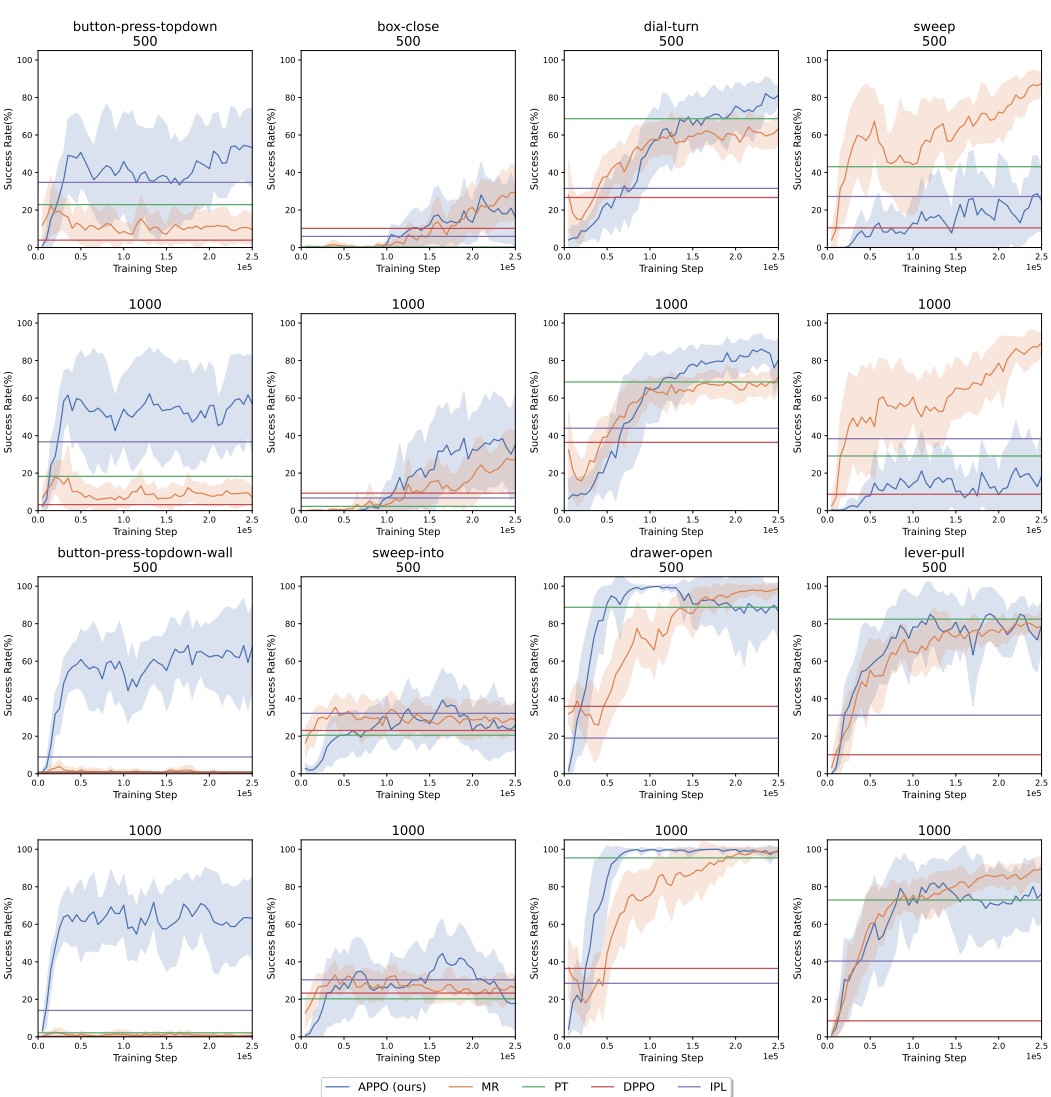

Figure 4: Learning Curves from the experiments in Table 1.

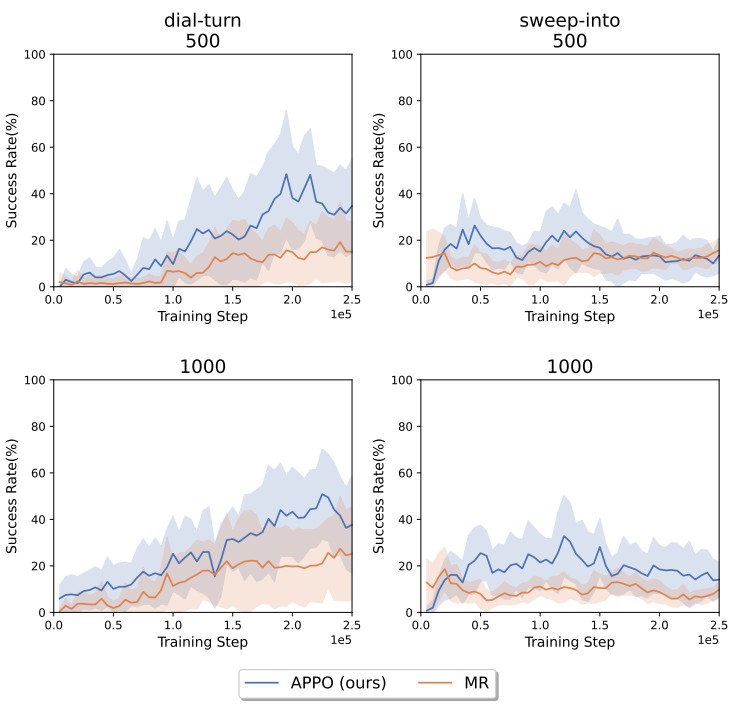

Figure 5: Learning Curves from the experiments in Table 2.

