# OpenReview forum: "Adversarial Policy Optimization for Offline Preference-based Reinforcement Learning"
_ICLR.cc/2025/Conference — ICLR 2025 Poster_

### Official Review · Reviewer_bzB5 · 2024-10-24

**Soundness:** 3
**Presentation:** 3
**Contribution:** 3
**Rating:** 8
**Confidence:** 2

**Summary:**

The authors propose a novel variant of the two-player formulation of PbRL, that allows the deviation of statistical bounds as well as a computational efficient implementation. The basic element of the proof is a novel sub-optimality decomposition. They also offer implementation details as well as an empirical comparison with other SoTA algorithms.

**Strengths:**

Theoretically sound approaches to PbRL are an important tool for researchers, because they allow conclusions beyond the empirically tested scenarios. Together with the also available empirical evaluation, this work becomes a significant contribution. This is further strengthened by the fact, that the algorithm is competitive to the SoTA. This is not expected for an algorithm derived from an mostly theoretical work.
The primary contribution of the work is an original contribution, which is embedded in a established framework.
Quality and Clarity are also good, with only minor limitations. Especially the formalization/introduction and the embedding into the available, related work are excellent. However, an overview table of related work, showing the available bounds, given assumptions & constraints would strength this further. This would enable the reader to directly pinpoint the variant considered by the authors, wrt. other works.

**Weaknesses:**

The most substantial weakness, are the repeated references to the appendix. The appendix should not be assumed to be available to every reader. In most cases, the explanations in the main paper are sufficient to follow the work without the appendix, but there are exceptions. E.g. the KL regularizer in alg 1&2 which is only requried for Alg.3, but this is not explained. In general, the authors should ensure that the paper is self contained. Stating a condensed version of what is written in the appendix (like for Theorem 4.1) is a good practice.

Furthermore, there are some strong assumptions, that should be discussed, like the fixed-length requirement or the Markovian reward. Both assumptions are correctly stated, but it is not clear if they are a substantial requirement, or just in place for enabling the formal proof.

Additionally, the experiments should be extended by additional ablation studies. As example, the impact of smaller/larger D_traj sets would be interesting (independent of the number of preferences). Another improvement would be adding a rank or critical distance plot for table 1, as a direct comparison with the SoTA is inconclusive. However, given that experiments are not the primary scope of the paper, these are only a smaller concerns.

One small typo: The results of baselines Oracle, PT, DPPO, and IPL are taken from Choi et al. (2024) - "Oracle" should likely by "MR"

**Questions:**

Can you please point me to the description of the size of $D_{traj}$ used in the experiments?

---

> ### Author Response · Authors · 2024-11-18
>
> We greatly appreciate the time and effort you have dedicated to reviewing our work and providing insightful feedback. Your feedback has been invaluable in helping us improve the clarity and presentation of our contributions. Below, we provide detailed responses to your comments.
>
> ### Completeness of Main Paper
>
> We appreciate your interest in the theoretical properties of Algorithm 1 and its foundational role in APPO (Algorithm 2). Due to space limitations, we concentrated on presenting the core ideas of Algorithm 1 in Section 3, while the detailed theoretical analysis and Algorithm 3 (a subroutine of Algorithm 1) were included in the Appendix. We recognize that a more comprehensive explanation of Algorithms 1 and 3 could improve clarity and are happy to revise the paper to provide a more self-contained presentation.
>
> The KL regularization ($\eta$) in the input list of Algorithm 3 is a typo, since $\eta$ is used in the policy update steps (Line 7 in Algorithm, Line 5 in Algorithm 2). Thank you for identifying this error.
>
> ### Reward Model
>
> Your request for clarification on the assumptions of our reward model is greatly valued, and we are grateful for the opportunity to provide clarity on these assumptions. The fixed-length (finite horizon) and Markovian reward assumptions provide a solid foundation for theoretical guarantees and practical implementations of our method. The Markovian reward assumption is standard and widely adopted in the field of PbRL [2-18], demonstrating its effectiveness in preference learning. While the fixed-length assumption facilitates our rigorous analysis, our method can be readily extended to the discounted, infinite horizon setting. Below, we elaborate on their significance:
>
> 1. **Fixed-length (Finite Horizon) Assumption:**
>     - The fixed-length (finite horizon) Markov Decision Process (MDP) assumption is essential for the theoretical analysis of PbRL. Similar to prior works on provably efficient offline PbRL algorithms [2,3,4] and online PbRL algorithms [5-11], our analysis relies on the finite horizon setting to relate the reward model error (Lemma E.2) to the suboptimality of the learned policy.
>     - Although our analysis is based on the finite horizon setting, this is a common assumption in the literature and does not constrain the practical applicability of our method. As discussed in Section 5, APPO can be straightforwardly implemented for the discounted, infinite horizon setting.
> 1. **Markovian Reward Assumption:**
>     - The Markovian reward assumption is crucial for both theoretical and practical considerations. From a theoretical perspective, the Markov property enables fundamental techniques in RL literature, such as the performance difference lemma and the Bellman equation. The lack of the Markovian property poses a significant challenge in offline PbRL, where the sub-optimality of learned policies must be bounded by model error. For this reason, the vast majority of studies in PbRL are based on the Markovian assumption [2-11]. To our best knowledge, there is only one work providing theoretical guarantee for non-Markovian preference [19], but it requires an online RL oracle to optimize policy with respect to the Markovian transformation of non-Markovian trajectory reward (Lemma 2.7 in [19]).
>     - Our practical implementation also requires a Markovian reward model, which is used to compute the loss functions. The Markovian reward assumption has been widely employed in empirical studies in PbRL [11-15]. Even the algorithms without explicit reward models assume implicit Markovian reward [16,17,18]. They have demonstrated Markovian reward models can successfully learn in complex tasks such as robotic control and games. This widespread use in the literature highlights that the Markovian reward assumption is standard and does not represent practical limitations of our approach.
>
> In conclusion, both the fixed-length (finite horizon) and Markovian reward assumptions are not constraints specific to our work but rather widely accepted and well-established model in the field of PbRL. Building on this solid theoretical foundation, APPO represents a significant advancement as the first offline PbRL algorithm to achieve both statistical and computational efficiency.

---

> ### Author Response · Authors · 2024-11-18
>
> ### Additional Experiments
>
> For a more extensive evaluation of our method, we conducted additional experiments varying the size of $D_{traj}$.  We compare APPO with MR, the most performant baseline according to our benchmark evaluation (Table 1). The tables below present the success rates of Meta-world dial-turn task with 1k preference feedback, where the size (number of transitions) of $D_{traj}$ varies from 50k to 300k . The performance of MR is unstable, as its success rate drops drastically for dataset sizes 100k and 200k. In contrast, APPO shows a gradual decrease in performance while dataset size decreases. This additional experiment and the result in Figure 2 demonstrate the robustness of APPO against dataset size.
>
> | size (x$10^5$) | $0.5$ | $1.0$ | $2.0$ | $3.0$ |
> | --- | --- | --- | --- | --- |
> | MR | $24.64 \pm 4.21$ | $7.84 \pm 4.22$ | $18.64 \pm 9.55$ | $69.44 \pm 4.70$ |
> | APPO | $21.68 \pm 7.08$ | $48.32 \pm 10.42$ | $63.12 \pm 7.57$ | $81.44 \pm 6.73$ |
>
> ### Experimental Details
>
> We used the size of $D_{traj}$ following the experimental protocol of [1], but the specific sizes were not described in the paper. Thank you for bringing this issue to our attention. we will ensure that the information is incorporated into the paper. The table below describes the sizes of $D_{traj}$ in Metaworld medium-replay datasets (BPT : button-press-topdown).
>
> | dataset | BPT | box-close | dial-turn | sweep | BPT-wall | sweep-into | drawer-open | lever-pull |
> | --- | --- | --- | --- | --- | --- | --- | --- | --- |
> | size (x$10^5$) | 1.0 | 8.0 | 3.0 | 7.0 | 1.5 | 1.0 | 1.0 | 3.0 |
>
> The results of MR is not taken from [1]. It is a reproduced result based on the implementation of [1]. The Oracle result (IQL trained with ground truth reward) is taken from [1].
>
> [1] Choi, H., Jung, S., Ahn, H. & Moon, T.. (2024). Listwise Reward Estimation for Offline Preference-based Reinforcement Learning. Proceedings of the 41st International Conference on Machine Learning, in Proceedings of Machine Learning Research 235:8651-8671
>
> [2] Wenhao Zhan, Masatoshi Uehara, Nathan Kallus, Jason D. Lee, and Wen Sun. Provable offline
> preference-based reinforcement learning. In The Twelfth International Conference on Learning
> Representations, 2024.
>
> [3] Banghua Zhu, Michael Jordan, and Jiantao Jiao. Principled reinforcement learning with human feedback from pairwise or k-wise comparisons. In International Conference on Machine Learning,
> pp. 43037–43067. PMLR, 2023.
>
> [4] Alizee Pace, Bernhard Sch\”olkopf, Gunnar R\”atsch, and Giorgia Ramponi. Preference elicitation for offline reinforcement learning. arXiv preprint arXiv:2406.18450, 2024.
>
> [5] Jonathan D Chang, Wenhao Shan, Owen Oertell, Kiante Brantley, Dipendra Misra, Jason D Lee, and Wen Sun. Dataset reset policy optimization for rlhf. arXiv preprint arXiv:2404.08495, 2024.
>
> [6] Ellen Novoseller, Yibing Wei, Yanan Sui, Yisong Yue, and Joel Burdick. Dueling posterior sampling for preference-based reinforcement learning. In Conference on Uncertainty in Artificial
> Intelligence, pp. 1029–1038. PMLR, 2020.
>
> [7] Yichong Xu, Ruosong Wang, Lin Yang, Aarti Singh, and Artur Dubrawski. Preference-based reinforcement learning with finite-time guarantees. Advances in Neural Information Processing
> Systems, 33:18784–18794, 2020.
>
> [8] Aadirupa Saha, Aldo Pacchiano, and Jonathan Lee. Dueling rl: Reinforcement learning with trajectory preferences. In International Conference on Artificial Intelligence and Statistics, pp. 6263–6289. PMLR, 2023.
>
> [9] Wenhao Zhan, Masatoshi Uehara, Wen Sun, and Jason D. Lee. Provable reward-agnostic preferencebased reinforcement learning. In The Twelfth International Conference on Learning Representations, 2024b.
>
> [10] Runzhe Wu and Wen Sun. Making rl with preference-based feedback efficient via randomization. arXiv preprint arXiv:2310.14554, 2023.
>
> [11] Yu Chen, Yihan Du, Pihe Hu, Siwei Wang, Desheng Wu, and Longbo Huang. Provably efficient iterated cvar reinforcement learning with function approximation and human feedback. In The Twelfth International Conference on Learning Representations, 2023.
>
> [12] Paul F Christiano, Jan Leike, Tom Brown, Miljan Martic, Shane Legg, and Dario Amodei. Deep
> reinforcement learning from human preferences. Advances in neural information processing systems, 30, 2017.
>
> [13] Ibarz, B., Leike, J., Pohlen, T., Irving, G., Legg, S., & Amodei, D. (2018). Reward learning from human preferences and demonstrations in atari. *Advances in neural information processing systems*, *31*.
>
> [14] Lee, K., Smith, L. M., & Abbeel, P. (2021, July). PEBBLE: Feedback-Efficient Interactive Reinforcement Learning via Relabeling Experience and Unsupervised Pre-training. In *International Conference on Machine Learning* (pp. 6152-6163). PMLR.

---

> > ### Author Response · Authors · 2024-11-18
> >
> > [15] Park, J., Seo, Y., Shin, J., Lee, H., Abbeel, P., & Lee, K. (2022). SURF: Semi-supervised reward learning with data augmentation for feedback-efficient preference-based reinforcement learning. *arXiv preprint arXiv:2203.10050*.
> >
> > [16] Changyeon Kim, Jongjin Park, Jinwoo Shin, Honglak Lee, Pieter Abbeel, and Kimin Lee. Preference transformer: Modeling human preferences using transformers for RL. In The Eleventh International Conference on Learning Representations, 2023.
> >
> > [17] Liu, R., Bai, F., Du, Y., & Yang, Y. (2022). Meta-reward-net: Implicitly differentiable reward learning for preference-based reinforcement learning. *Advances in Neural Information Processing Systems*, *35*, 22270-22284.
> >
> > [18] Joey Hejna and Dorsa Sadigh. Inverse preference learning: Preference-based rl without a reward function. Advances in Neural Information Processing Systems, 36, 2024.
> >
> > [19] Gokul Swamy, Christoph Dann, Rahul Kidambi, Steven Wu, and Alekh Agarwal. A minimaximalist approach to reinforcement learning from human feedback. In Forty-first International Conference on Machine Learning, 2024.

---

> ### Author Response · Authors · 2024-11-21
>
> We have uploaded the revised paper incorporating your feedback, which has been instrumental in improving and refining our work. We sincerely thank you for your detailed review and insightful comments.
>
> - In Section 3.2, we corrected the typo regarding Algorithms 1 and 3 (KL regularization $\eta$) and enhanced the description of Algorithm 3 to make its role clearer to readers.
> - Table 1 now presents the average ranks of algorithms, providing a summary of their relative performance.
> - We conducted an additional experiment to evaluate the effect of $|D_{traj}|$. The table below presents the result on Meta-world medium replay sweep-into dataset, with 1000 preference feedback. The success rate of APPO improves with larger datasets, whereas MR does not exhibit a clear correlation. Please refer to **Appendix F.2** for the corresponding plots.
>
> | size (x$10^5$) | $0.5$ | $1.0$ | $1.5$ | $2.0$ |
> | --- | --- | --- | --- | --- |
> | MR | $3.28\pm1.20$ | $26.00\pm5.53$ | $15.44\pm5.14$ | $21.28\pm8.37$ |
> | APPO | $3.20\pm1.13$  | $18.16\pm11.14$ | $38.72\pm14.97$ | $55.68\pm11.16$ |

---

### Official Review · Reviewer_FTyb · 2024-10-27

**Soundness:** 3
**Presentation:** 2
**Contribution:** 2
**Rating:** 8
**Confidence:** 2

**Summary:**

This work explores the innovative realm of PbRL, addressing the challenges of ensuring conservatism under uncertainty. APPO is designed to ensure conservatism in PbRL without the need for explicit construction of intractable confidence sets. This is achieved by framing PbRL as a two-player game between a policy and a model, which allows for a tractable enforcement of conservatism. Experimental results show that APPO performs comparably to existing state-of-the-art algorithms in continuous control tasks．

**Strengths:**

APPO is designed to optimize the learning process in preference-based reinforcement learning (PbRL) by effectively utilizing preference feedback, which allows for faster convergence and improved sample efficiency compared to traditional methods. The proposed method can be integrated with any standard unconstrained reinforcement learning algorithm, making it versatile and applicable across various domains. Additionally, the paper provides theoretical bounds on the sample complexity of the proposed method.

**Weaknesses:**

The performance of APPO is sensitive to the choice of the conservatism regularizer coefficient (λ). While the algorithm can learn with a range of λ values, improper tuning can lead to suboptimal performance and stability issues, which may require additional effort in hyperparameter optimization. Moreover, the theoretical guarantees of APPO rely on standard assumptions regarding function approximation and bounded trajectory concentrability. If these assumptions do not hold in certain environments, the performance and reliability of the algorithm may be compromised.

**Questions:**

1.What is the impact of the conservatism regularizer coefficient (λ) on the performance of APPO? How can an appropriate λ value be selected?

2.How does APPO compare to other IQL-based algorithms in terms of hyperparameter tuning advantages and disadvantages?

3.What happens to the performance of APPO if these assumptions do not hold in certain environments?

4.What are the scalability concerns of APPO in high-dimensional state or action spaces? Are there any issues related to computational complexity?

5.In practical applications, how does APPO cope with sparse or non-representative preference data?

---

> ### Author Response · Authors · 2024-11-18
>
> Thank you for dedicating your time and expertise to reviewing our work and for offering constructive feedback. Below, we provide detailed responses to address your comments and questions.
>
> 1. **Impact of Conservatism Regularizer:**
>     - The effect of the conservatism regularizer is demonstrated in Section 6 (Figure 1), and the experiments in Table 1 share the same $\lambda$ value. These results show that APPO is robust to change in $\lambda$.
>     - The conservatism regularizer balances conservatism and model error. Without conservatism, errors may amplify due to distributional shift [1], while excessive conservatism leads to a large bias in value estimation.
>     - When tuning the value of $\lambda$, we suggest searching within the range $\lambda \leq 1$. Alternatively, $\lambda$ can be optimized during training by specifying a target value and applying gradient descent. However, this approach still requires tuning the target $\lambda$ value.
>
>
> 2. **Advantage in Hyperparameter Tuning:**
>     - APPO has only one key algorithmic hyperparameter, aside from standard hyperparameters in deep learning, such as learning rates. This simplicity is an advantage over IQL-based algorithms, which require tuning at least two key hyperparameters, as described below. While the reward model may require hyperparameter tuning depending on the specific implementation, our experiments show that a simple feed-forward network trained via maximum likelihood estimation achieves strong performance.
>     - In contrast, the IQL-based algorithms involve additional hyperparameters, such as the expectile regression parameter ($\tau$) and the inverse temperature for advantage weighted regression ($\beta$) [2]. IPL [20] introduces another regularization parameter (referred to as \lambda therein). This makes APPO’s single hyperparameter an advantage in terms of simplicity. One potential downside is the entropy regularizer in APPO ($\alpha$ in equation 11), which is not present in IQL. However, we followed the standard recipe from soft actor-critic [3] without additional tuning, so it was not counted as a tuning parameter in our experiments.
>
> 3. **Discussion on Assumptions:**
>     - The realizability assumptions (Assumptions 1,2,3) are not practical concerns when using powerful function approximators such as neural networks.
>     - The trajectory concentrability (Assumption 4) ensures the dataset contains high-quality trajectories. If violated, performance could degrade, as in the case of all offline PbRL algorithms [4-7]. However, APPO demonstrated consistent performance across diverse environments in Section 6, supporting its robustness.
>     - The Markovian reward assumption (equation 1) has been widely adopted in the PbRL literature [4-20]. Even algorithms without explicit reward models often assume implicit Markovian reward [18,19,20]. These studies have shown that Markovian rewards enable successful learning in complex tasks such as robotic control and games, highlighting that this assumption is standard and does not impose practical limitations.
>
> 4. **Scalability and Computational Complexity:**
>     - The practical implementation of APPO in Section 5 uses parameterized value functions and policies trained via standard gradient descent. The loss function for policy (equation 11) is similar to standard actor-critic, and the loss functions for value functions (equations 9,10) require a comparable computational cost compared to standard TD learning methods. As a result, APPO can scale effectively to large datasets or high-dimensional state/action spaces, similar to existing deep RL algorithms. In our computational setup, both MR (IQL) and APPO require about 4 hours to take 250k gradient steps.
>     - Training a reward model is relatively inexpensive compared to training a policy. For instance, on our computational setup, training a reward model with 1,000 preference feedback samples takes less than 30 seconds.
>
> 5. **Sparse or Non-representative Preference:**
>     - The preference data used in our experiments is sparse, as preference feedback is collected from randomly sampled trajectory segment pairs within the dataset. Given the number of preference feedbacks ($500$ or $1,000$) is much smaller than the dataset size ($>10^5$), dense preference data is highly unlikely.
>     - We kindly request clarification on the meaning of 'non-representative preference' to ensure a more precise and thorough discussion in response to your feedback.

---

> > ### Author Response · Authors · 2024-11-18
> >
> > [1] Levine, S., Kumar, A., Tucker, G., & Fu, J. (2020). Offline reinforcement learning: Tutorial, review, and perspectives on open problems. *arXiv preprint arXiv:2005.01643*.
> >
> > [2] Kostrikov, I., Nair, A., & Levine, S. Offline Reinforcement Learning with Implicit Q-Learning. In *International Conference on Learning Representations*.
> >
> > [3] Tuomas Haarnoja, Aurick Zhou, Pieter Abbeel, and Sergey Levine. Soft actor-critic: Off-policy
> > maximum entropy deep reinforcement learning with a stochastic actor. In International conference on machine learning, pp. 1861–1870. PMLR, 2018.
> >
> > [4] Wenhao Zhan, Masatoshi Uehara, Nathan Kallus, Jason D. Lee, and Wen Sun. Provable offline
> > preference-based reinforcement learning. In The Twelfth International Conference on Learning
> > Representations, 2024.
> >
> > [5] Banghua Zhu, Michael Jordan, and Jiantao Jiao. Principled reinforcement learning with human feedback from pairwise or k-wise comparisons. In International Conference on Machine Learning,
> > pp. 43037–43067. PMLR, 2023.
> >
> > [6] Alizee Pace, Bernhard Sch\”olkopf, Gunnar R\”atsch, and Giorgia Ramponi. Preference elicitation for offline reinforcement learning. arXiv preprint arXiv:2406.18450, 2024.
> >
> > [7] Jonathan D Chang, Wenhao Shan, Owen Oertell, Kiante Brantley, Dipendra Misra, Jason D Lee, and Wen Sun. Dataset reset policy optimization for rlhf. arXiv preprint arXiv:2404.08495, 2024.
> >
> > [8] Ellen Novoseller, Yibing Wei, Yanan Sui, Yisong Yue, and Joel Burdick. Dueling posterior sampling for preference-based reinforcement learning. In Conference on Uncertainty in Artificial
> > Intelligence, pp. 1029–1038. PMLR, 2020.
> >
> > [9] Yichong Xu, Ruosong Wang, Lin Yang, Aarti Singh, and Artur Dubrawski. Preference-based reinforcement learning with finite-time guarantees. Advances in Neural Information Processing
> > Systems, 33:18784–18794, 2020.
> >
> > [10] Aadirupa Saha, Aldo Pacchiano, and Jonathan Lee. Dueling rl: Reinforcement learning with trajectory preferences. In International Conference on Artificial Intelligence and Statistics, pp. 6263–6289. PMLR, 2023.
> >
> > [11] Wenhao Zhan, Masatoshi Uehara, Wen Sun, and Jason D. Lee. Provable reward-agnostic preferencebased reinforcement learning. In The Twelfth International Conference on Learning Representations, 2024b.
> >
> > [12] Runzhe Wu and Wen Sun. Making rl with preference-based feedback efficient via randomization. arXiv preprint arXiv:2310.14554, 2023.
> >
> > [13] Yu Chen, Yihan Du, Pihe Hu, Siwei Wang, Desheng Wu, and Longbo Huang. Provably efficient iterated cvar reinforcement learning with function approximation and human feedback. In The Twelfth International Conference on Learning Representations, 2023.
> >
> > [14] Paul F Christiano, Jan Leike, Tom Brown, Miljan Martic, Shane Legg, and Dario Amodei. Deep
> > reinforcement learning from human preferences. Advances in neural information processing systems, 30, 2017.
> >
> > [15] Ibarz, B., Leike, J., Pohlen, T., Irving, G., Legg, S., & Amodei, D. (2018). Reward learning from human preferences and demonstrations in atari. *Advances in neural information processing systems*, *31*.
> >
> > [16] Lee, K., Smith, L. M., & Abbeel, P. (2021, July). PEBBLE: Feedback-Efficient Interactive Reinforcement Learning via Relabeling Experience and Unsupervised Pre-training. In *International Conference on Machine Learning* (pp. 6152-6163). PMLR.
> >
> > [17] Park, J., Seo, Y., Shin, J., Lee, H., Abbeel, P., & Lee, K. (2022). SURF: Semi-supervised reward learning with data augmentation for feedback-efficient preference-based reinforcement learning. *arXiv preprint arXiv:2203.10050*.
> >
> > [18] Changyeon Kim, Jongjin Park, Jinwoo Shin, Honglak Lee, Pieter Abbeel, and Kimin Lee. Preference transformer: Modeling human preferences using transformers for RL. In The Eleventh
> > International Conference on Learning Representations, 2023.
> >
> > [19] Liu, R., Bai, F., Du, Y., & Yang, Y. (2022). Meta-reward-net: Implicitly differentiable reward learning for preference-based reinforcement learning. *Advances in Neural Information Processing Systems*, *35*, 22270-22284.
> >
> > [20] Joey Hejna and Dorsa Sadigh. Inverse preference learning: Preference-based rl without a reward function. Advances in Neural Information Processing Systems, 36, 2024.

---

> > > ### Comment · Reviewer_FTyb · 2024-11-21
> > >
> > > Thanks for your response, I will improve my rating.

---

### Official Review · Reviewer_bKme · 2024-11-03

**Soundness:** 3
**Presentation:** 3
**Contribution:** 3
**Rating:** 6
**Confidence:** 3

**Summary:**

This paper studies offline PbRL. The authors propose APPO, which frames PbRL as a two-player game between the policy and an adversarial reward model. This algorithm enforces pessimism without explicit confidence sets, making the method more computationally tractable than the existing literature. The paper further provides theoretical guarantees, demonstrating that APPO achieves strong sample complexity bounds under standard assumptions on function approximation and trajectory concentrability. Experimental results on continuous control tasks (in Meta-world) show that APPO achieves performance comparable to state-of-the-art methods.

**Strengths:**

1. Algorithmic novelty: The Q function estimation part in Algorithm 2 is interesting and novel.

2. Valid theoretical guarantee: The authors provide a rigorous sample complexity analysis under standard assumptions.

3. Qualified empirical performance: The experiments show that the practical implementation of APPO is a competitive empirical algorithm. This part is lacked in many existing works.

**Weaknesses:**

1. Limited novelty: the essential idea is not very novel. In standard RL, there have been works to use two-player games to get rid of confidence sets (e.g., 'Adversarially Trained Actor Critic for Offline Reinforcement Learning' by Ching-An Cheng) and this work is very similar to this line of work. In addition, the authors should highlight new analysis techniques in the paper, if any.


2. Complicated algorithm: although Algorithm 2 doesn't need confidence sets, the algorithm is still not computationally efficient due to Line 4. Typically, most of the existing works think a method computationally efficient if it can be solved with least squares oracles and MLE oracles. However, I think Line 4 in Algorithm 2 cannot be solved with these oracles.

**Questions:**

1. The authors claim that using Lagrangian multipliers to get rid of the confidence sets is not applicable here (Line 204-207). I am not quite convinced because such an approach is effective in standard RL and I hope the authors can elaborate more.


2. For Algorithm 2, can we just replace the true environment in Algorithm 1 with the estimated environment $\widehat{P}$ (adding some regularizers to quantify the uncertainty of $\widehat{P}$ at the same time)? Then the resulted algorithm could be less complicated.

---

> ### Author Response · Authors · 2024-11-18
>
> We thank you for your time and effort in reviewing our paper and for providing feedback. Below, we address your comments and questions in detail, and we hope our responses clarify the key contributions.
>
> ### Adversarial Training
>
> We respectfully disagree with the assertion that our work lacks novelty and are glad to clarify its contributions. Our proposed method APPO is a novel adaptation of the two-player game framework tailored to the unique challenges of Preference-based Reinforcement Learning (PbRL) — which we elaborate in the paper and below. While related frameworks have been employed in standard RL, our approach departs significantly in its formulation, analysis, and application, addressing the complexities of PbRL that are not encountered in standard RL settings.
>
> 1. **Distinction from Model-based Adversarial Training in Standard RL:**
>     - In standard RL, model-based adversarial training [1,3,4] formulates a two-player game between the transition model and the policy:
> $$
> \max_{\pi} \min_{P \in \mathcal{P}} J(\pi, P),
> $$
> where $ \mathcal{P} $ is the confidence set of transition models, and $J$ is the expected return under policy $\pi$ and transition model $P$. This approach is computationally intractable due to the confidence set. As a workaround, previous works [1,3] replace the constraint with a regularization term but sacrifice theoretical guarantees for practical feasibility.
>     - **In contrast, APPO provides sample complexity bounds for the regularized optimization framework, achieving both statistical and computational efficiency.** This distinguishes our work by offering both theoretical rigor and practical applicability.
>
> 2. **Distinction from Bellman-consistency-based Adversarial Training:**
>     - Bellman-consistency-based methods [2,7] frame the two-player game between the value function and the policy. While these methods provide performance guarantees for the regularized optimization form, their analysis differs fundamentally from ours.
>     - **Our work differs in being model-based:** APPO regularizes the deviation of the reward model (and the induced value function) from the estimated reward model. Conversely, Bellman-consistency-based methods are model-free and focus on regularizing the Bellman error of the value function. Consequently, our sub-optimality decomposition (Section 4 and Appendix D) bounds policy sub-optimality using model estimation error, offering a fundamentally different analytical approach.
>
> 3. **Unique Challenges in PbRL:**
>     - PbRL introduces trajectory-based feedback, making it significantly more challenging than standard RL. For example, in offline learning:
>     - Standard RL requires step-wise policy concentrability, which is sufficient to bound sample complexity [5].
>     - PbRL demands trajectory-wise policy concentrability, resulting in a polynomial dependence on this parameter in the sample complexity bounds (Theorem 2, Theorem 3 in [6]).
>     - Due to these challenges, existing offline PbRL algorithms are computationally intractable, and standard RL analyses fail to provide guarantees in the PbRL setting.
>     - **APPO successfully addresses these challenges using our adversarial training framework, complemented by a novel analysis that bridges theoretical guarantees and practical feasibility.**
>
> In conclusion, APPO is a significant contribution that advances the state-of-the-art in PbRL by addressing its unique challenges through a novel two-player game formulation, enhanced theoretical analysis, and practical algorithm design. Thank you for the opportunity to clarify our contributions.

---

> > ### Author Response · Authors · 2024-11-18
> >
> > ### Computational Complexity
> >
> > We respectfully clarify that the computational complexity of Line 4 in Algorithm 2 is comparable to that of widely used least squares or maximum likelihood estimation (MLE) in the context of general function approximation. In this regime, least squares and MLE are typically non-convex and non-linear optimization problems. For example, when the reward model is parameterized by a neural network, finding the maximum likelihood parameters inherently involves solving a non-convex optimization problem. Similarly, Line 4 of Algorithm 2 requires solving a non-convex optimization, making its computational complexity analogous to that of least squares or MLE in the general function approximation setting. It is important to note that any algorithm relying on oracles in this setting would share at least the same order of complexity as Line 4, making this a standard requirement rather than a unique challenge of our approach.
> >
> > Furthermore, modern deep learning techniques have made such non-convex optimization problems both practical and efficient. For neural function approximation, these problems can be solved using widely available deep learning libraries and frameworks, ensuring that the computational demands of Line 4 are manageable in practice. Consequently, our algorithm is computationally feasible and does not impose additional challenges beyond those encountered in standard approaches involving neural networks or other general function approximations. We believe this perspective underscores that the computational requirements of our method align with the state of the art and remain practical.
> >
> > ### Role of Algorithm 3 as A Subroutine of Algorithm 1
> >
> > Regarding the second question, the true environment is necessary for the performance guarantee of Algorithm 1 (Theorem C.2). If the policy evaluation subroutine (Algorithm 3) uses the estimated transition, the Monte Carlo estimation becomes biased, invalidating the error bound established in Lemma B.1. Additionally, we highlight that Algorithm 1 serves as a foundational building block for our main algorithm APPO (Algorithm 2). By leveraging our reparameterization technique, APPO eliminates the need for the policy evaluation subroutine (Algorithm 3) while maintaining strong statistical guarantees.
> >
> > [1] Marc Rigter, Bruno Lacerda, and Nick Hawes. Rambo-rl: Robust adversarial model-based offline reinforcement learning. Advances in neural information processing systems, 35:16082–16097, 2022.
> >
> > [2] Ching-An Cheng, Tengyang Xie, Nan Jiang, and Alekh Agarwal. Adversarially trained actor critic
> > for offline reinforcement learning. In International Conference on Machine Learning, pp. 3852–
> > 3878. PMLR, 2022.
> >
> > [3] Mohak Bhardwaj, Tengyang Xie, Byron Boots, Nan Jiang, and Ching-An Cheng. Adversarial model for offline reinforcement learning. Advances in Neural Information Processing Systems, 36, 2024.
> >
> > [4] Aravind Rajeswaran, Igor Mordatch, and Vikash Kumar. A game theoretic framework for model
> > based reinforcement learning. In International conference on machine learning, pp. 7953–7963.
> > PMLR, 2020.
> >
> > [5] Masatoshi Uehara and Wen Sun. Pessimistic model-based offline reinforcement learning under partial coverage. In International Conference on Learning Representations, 2022.
> >
> > [6] Wenhao Zhan, Masatoshi Uehara, Nathan Kallus, Jason D. Lee, and Wen Sun. Provable offline
> > preference-based reinforcement learning. In The Twelfth International Conference on Learning
> > Representations, 2024.
> >
> > [7] Tengyang Xie, Ching-An Cheng, Nan Jiang, Paul Mineiro, and Alekh Agarwal. Bellman-consistent pessimism for offline reinforcement learning. Advances in Neural Information Processing Systems, 34:6683–6694, 2021a.

---

> > > ### Comment · Reviewer_bKme · 2024-11-19
> > >
> > > Hi,
> > >
> > > Thanks for the feedback! The authors have addressed my concerns well and thus I have improved my rating.

---

### Official Review · Reviewer_DhLA · 2024-11-04

**Soundness:** 4
**Presentation:** 3
**Contribution:** 3
**Rating:** 6
**Confidence:** 3

**Summary:**

This paper introduces APPO, a novel algorithm for offline PbRL that utilizes a two-player game formulation to produce a robust policy less prone to model errors. By avoiding explicit confidence sets, APPO achieves computational and statistical efficiency. The paper derives a sample complexity bound for APPO under standard assumptions and presents experimental results demonstrating performance comparable to SOTA offline PbRL methods on continuous control tasks.

**Strengths:**

This paper is well-motivated and organized. The proposed approach is theoretically sound and solid. The authors derive sample complexity bounds without relying on explicit confidence sets, offering statistical efficiency and practical applicability. The approach is evaluated on continuous control tasks, with results showing that it performs on par with or surpasses SOTA baselines.

**Weaknesses:**

Please see Questions.

**Questions:**

1. The paper only evaluates APPO on a medium-replay dataset from Metaworld. It would be beneficial to evaluate APPO on a wider range of datasets to assess its generalizability and robustness.

2. Previous work on offline PbRL, such as CPL [1], directly learns a policy without RL, as noted in the related work section. Using a supervised learning approach, CPL achieves comparable performance with SOTA baselines while significantly reducing computational complexity. How does APPO compare to CPL, and what are the advantages of using APPO over CPL?

3. Could the authors discuss the limitations of APPO?

[1] Joey Hejna, Rafael Rafailov, Harshit Sikchi, Chelsea Finn, Scott Niekum, W. Bradley Knox, and Dorsa Sadigh. Contrastive preference learning: Learning from human feedback without reinforcement learning. In The Twelfth International Conference on Learning Representations, 2024.

---

> ### Author Response · Authors · 2024-11-20
>
> We deeply appreciate the time and effort you have invested in reviewing our paper and providing thoughtful feedback. We hope our response clarifies your questions.
>
> 1. **Experiments with Additional Datasets:**
>
>     To further demonstrate the generalization capability of APPO, we collected Meta-world [1] medium-expert datasets following the approaches of prior works [2, 3]. As a baseline, we selected MR since it was the most performant baseline evidenced in our benchmark experiment (Table 1). The hyperparameters of APPO and MR are identical to the values used in Section 6.
>
> | # of feedback | $500$ |  | $1000$ |  |
> | --- | --- | --- | --- | --- |
> | dataset | dial-turn | sweep-into | dial-turn | sweep-into |
> | MR | $15.80\pm  12.73$ | $14.32\pm3.39$ | $26.08\pm 18.78$ | $8.48\pm1.92$ |
> | APPO | $32.40\pm13.56$ | $12.80\pm5.35$ | $39.20\pm15.69$  | $14.56\pm6.25$ |
>
> The table above shows the success rates on the Meta-world medium-expert datasets. APPO outperforms or performs on par with MR. The result exhibits the generalization capability and robustness of APPO in a wide range of datasets.
>
> We added this experiment in **Appendix F.1**. Please refer to the revised paper for the learning curves of the experiment. Thank you for your constructive feedback, which has helped us refine and improve our work.
>
> **Details on Meta-world medium-expert dataset:**
>
> - Each dataset contains trajectories from five sources: (1) the expert policy, (2) expert policies for randomized variants and goals of the task, (3) expert policies for different tasks, (4) a random policy, and (5) an $\epsilon$-greedy expert policy that takes greedy actions with a $50$% probability. These trajectories are included in the dataset in proportions of $1 : 1 : 2 : 4 : 4$, respectively. Additionally, standard Gaussian noise was added to the actions of each policy.
> - The dataset sizes match those of the medium-replay dataset in Table 3.
> - Preference feedback is labeled as described in Section 6.
>
> 2. **Comparison with CPL [4]:**
>     - The primary differences lie in the preference models and learning objectives. APPO assumes a standard trajectory-return-based preference model and aims to maximize cumulative reward, whereas CPL employs a regret-based preference model with an entropy-regularized cumulative reward as its learning objective.
>     - These differences lead to distinct algorithmic approaches: APPO uses a value-learning method similar to actor-critic, while CPL relies on optimizing a supervised learning objective.
>     - A key advantage of APPO over CPL is its provable sample complexity bound, as CPL does not provide finite-sample performance guarantees. However, direct comparison with CPL is difficult as our primary contribution lies in the theoretical analysis, while CPL is designed to achieve computational efficiency in empirical scenarios.
>
> 3. **Limitations of APPO:**
>
>     APPO is a model-based algorithm, as its theoretical analysis relies on the MLE error bound of the reward model. To the best of our knowledge, existing provably efficient offline PbRL algorithms are model-based. Although training the reward model is computationally inexpensive compared to policy training (training a reward model takes less than 30 seconds, while policy training requires over 3 hours in our setting), eliminating the explicit reward model remains an intriguing theoretical challenge.
>
> [1] Tianhe Yu, Deirdre Quillen, Zhanpeng He, Ryan Julian, Karol Hausman, Chelsea Finn, and Sergey Levine. Meta-world: A benchmark and evaluation for multi-task and meta reinforcement learning. In Conference on robot learning, pp. 1094–1100. PMLR, 2020.
>
> [2] Choi, H., Jung, S., Ahn, H. & Moon, T.. (2024). Listwise Reward Estimation for Offline Preference-based Reinforcement Learning. Proceedings of the 41st International Conference on Machine Learning, in Proceedings of Machine Learning Research 235:8651-8671
>
> [3] Joey Hejna and Dorsa Sadigh. Inverse preference learning: Preference-based rl without a reward function. Advances in Neural Information Processing Systems, 36, 2024.
>
> [4] Hejna, J., Rafailov, R., Sikchi, H., Finn, C., Niekum, S., Knox, W. B., & Sadigh, D. Contrastive Preference Learning: Learning from Human Feedback without Reinforcement Learning. In *The Twelfth International Conference on Learning Representations*.

---

### Meta-Review · Area_Chair_p2VK · 2024-12-19

**Metareview:**

The paper studied offline preference-based learning and proposed an algorithm that is more computationally tractable than prior works. Theoretically, the paper extends ideas from pessimistic offline RL literature to offline preference learning. From the empirical side, the proposed algorithm achieved good performance on the standard continuous control tasks.

**Additional Comments On Reviewer Discussion:**

Reviewers in general are positive about this paper. Before the rebuttal, the reviewers raised concerns on the novelty of the theoretical analysis and additional experiments. During the rebuttal, the authors provided additional experiments, additional clarification and justification of why the proposed approach is novel and is different from prior offline RL methods, and additional clarification of why their proposed approach is computationally tractable. The authors rebuttal addressed these concerns from reviewers and made reviewers increased scores.

---

### Decision · Program_Chairs · 2025-01-22

Accept (Poster)